# QUERY EFFICIENT BLACK-BOX ADVERSARIAL ATTACK WITH AUTOMATIC REGION SELECTION

## ABSTRACT

Deep neural networks (DNNs) have been shown to be vulnerable to black-box attacks in which small perturbations are added to input images without accessing any internal information of the model. However, current black-box adversarial attack methods are limited to attacks on entire regions, pixel-wise sparse attacks, or region-wise attacks. In this paper, we investigate region-wise adversarial attacks in the black-box setting, using automatic region selection and controllable imperceptibility. Technically, we formulate the problem as an optimization problem with $\ell_0^{\mathcal{G}}$ and $\ell_\infty$ constraints. Here, $\ell_0^{\mathcal{G}}$ represents structured sparsity defined on one collection of groups $\mathcal{G}$, which can automatically detect the regions that need to be perturbed. We solve the problem using the algorithm of natural evolution strategies with search gradients. If $\mathcal{G}$ is non-overlapping, we provide a closed-form solution to the first-order Taylor approximation of the objective function with the search gradient having $\ell_0^{\mathcal{G}}$ and $\ell_\infty$ constraints (FTAS$\ell_{0+\infty}^{\mathcal{G}}$). If $\mathcal{G}$ is overlapping, we provide an approximate solution to FTAS$\ell_{0+\infty}^{\mathcal{G}}$ due to its NP-hard nature, using greedy selection on the collection of groups $\mathcal{G}$. Our method consists of multiple updates with the closed-form/approximate solution to FTAS$\ell_{0+\infty}^{\mathcal{G}}$. We provide the convergence analysis of the solution under standard assumptions. Our experimental results on different datasets indicate that we require fewer perturbations compared to global-region attacks, fewer queries compared to region-wise attacks, and better interpretability into vulnerable regions which is not possible with pixel-wise sparse attacks.

## 1 INTRODUCTION

Deep neural networks (DNNs) have gained significant attention and are widely adopted in various applications, including computer vision (He et al., 2017; 2016), security systems (Kang & Kang, 2016; Xibilia et al., 2020), natural language processing (Bahdanau et al., 2016; Joshi et al., 2019), and autonomous driving (Bojarski et al., 2016; Levinson et al., 2011; Xiong et al., 2019). However, extensive experiments have revealed that DNNs are susceptible to adversarial attacks, where well-designed small perturbations can deceive the models (Cai et al., 2021; Cheng et al., 2018; Su et al., 2019; Zhao et al., 2019). The methods of adversarial attack can be classified into two main categories: white-box and black-box attacks. White-box attacks assume access to the target model, enabling the attacker to directly update adversarial examples using the gradients of the model (Dong et al., 2020; Fan et al., 2020; Kazemi et al., 2023; Zhu et al., 2021). However, in numerous real-world scenarios, models are inaccessible, rendering gradient calculations impossible. In such situations, black-box attackers aim to approximate gradients by querying the target network to obtain output predictions for input samples. This paper focuses on discussing black-box attacks.

Currently, there is a considerable amount of research dedicated to studying the adversarial vulnerability of networks in the black-box setting. The majority of these studies primarily focus on developing attacks (Ilyas et al., 2018a;b; Tu et al., 2019; Zhao et al., 2020) that target entire regions. Specifically, ZO-NGD (Zhao et al., 2020), which imposes an $\ell_\infty$ constraint, incorporates the zeroth-order gradient estimation technique and the second-order natural gradient to generate imperceptible perturbations on the entire image. (Ilyas et al., 2018a) proposed a method based on Natural Evolutionary Strategies to estimate the gradient under $\ell_\infty$ constraint. In the next year, they further proposed based on prior information to improve the query efficiency (Ilyas et al., 2018b). However, since global perturbation alters the statistical characteristics of the entire image, it may in-

Figure 1: A demonstration of adversarial examples and the corresponding perturbations generated by our method, Patch-RS, and Square Attack. Our method effectively identifies the region containing the target within the perturbed image and generates perturbations that align better with the target's location. Patch-RS drew a conspicuous patch on the image. Square Attack with $\ell_\infty$ constraint does not have any image structure, and the perturbation is obvious.

troduce abnormal visual effects. These effects have the potential to be detected not only by defense mechanisms but also by human observers.

In addition to zeroth-order optimization to estimate gradient, there is also a heuristic search method in the black-box attack. For instance, Square Attack (Andriushchenko et al., 2020) is based on a random search scheme, which selects local square updates at random locations so that the perturbation in each iteration is approximately located at the boundary of the feasible solution. But it can not be ignored that it will cause more noise in the large region even the entire image, which potentially makes the perturbations more visually apparent. Parsimonious Attack (Moon et al., 2019) divides the image into some blocks according to some coarse grid. Then it performs a local heuristic search in a low-dimensional space among the vertices of the $\ell_\infty$ ball. Differently, pixel-wise sparse attacks (de Vazelhes et al., 2022; Croce & Hein, 2019; Tian et al., 2022) focus on identifying pixels that contribute significantly to the attack and independently applying perturbations to these pixels. Since natural images often exhibit a local smoothness property from a statistical perspective, the addition of perturbations usually disrupts this local smoothness property, rendering the perturbations more easily detectable by defense mechanisms.

Recently, region-wise attacks have been proposed, allowing attackers to exploit vulnerabilities in specific regions or input areas. By understanding the model's behavior with specific regions, attackers can design targeted perturbations to manipulate the model's predictions in a desired manner. However, existing black-box region-wise attacks usually achieve bad performance in terms of success attack rate due to the nature of the black-box setting and unreasonable regional selection. For instance, Croce et al. (Croce et al., 2022) perturb only the 2-pixel wide edges of the original image, or add a patch at arbitrary locations based on a heuristic random search. Therefore, how to find the region that can highly improve the success attack rate has become a crucial problem in region-wise attacks.

To address this challenge, we propose an approach for black-box attacks, which automatically detects the relevant regions based on a reliable standard instead of a fixed region or heuristics. As shown in Fig. 1, we find that FTAS$\ell_{0+\infty}^{\mathcal{G}}$ produces perturbations that fit the target, outperforming Patch-RS and Square Attack methods in terms of perturbation quality and suitability. Different from heuristic-based approaches, we formulate the problem as an optimization problem with $\ell_0^{\mathcal{G}}$ and $\ell_\infty$ constraints technically. Here, $\ell_0^{\mathcal{G}}$ represents structured sparsity defined on one collection of groups $\mathcal{G}$, which can automatically detect the regions that need to be perturbed. We solve the problem using the algorithm of natural evolution strategies with search gradients. Specifically, if $\mathcal{G}$ is non-overlapping, we provide a closed-form solution to the first-order Taylor approximation of the objective function with the search gradient having $\ell_0^{\mathcal{G}}$ and $\ell_\infty$ constraints (FTAS$\ell_{0+\infty}^{\mathcal{G}}$). If $\mathcal{G}$ is overlapping, we provide an approximate solution to FTAS$\ell_{0+\infty}^{\mathcal{G}}$ due to its NP-hard nature, using greedy selection on the collection of groups $\mathcal{G}$. In addition, we provide a geometric convergence rate in Theorem 2 under the standard assumptions. We conduct experiments on different datasets that demonstrate the proposed method requires fewer perturbations and queries compared to global-region and region-wise attacks respectively, and provide better interpretability and insights into vulnerable regions than pixel-wise sparse attacks.

## 2 ADVERSARIAL ATTACK WITH AUTOMATICAL REGION DETECTION

In this section, we begin with a concise overview of adversarial attacks. Subsequently, we present our novel framework for adversarial attacks, which incorporates an automated region detection mechanism. A visual comparison between ours and heuristic methods is shown in Fig. 2.

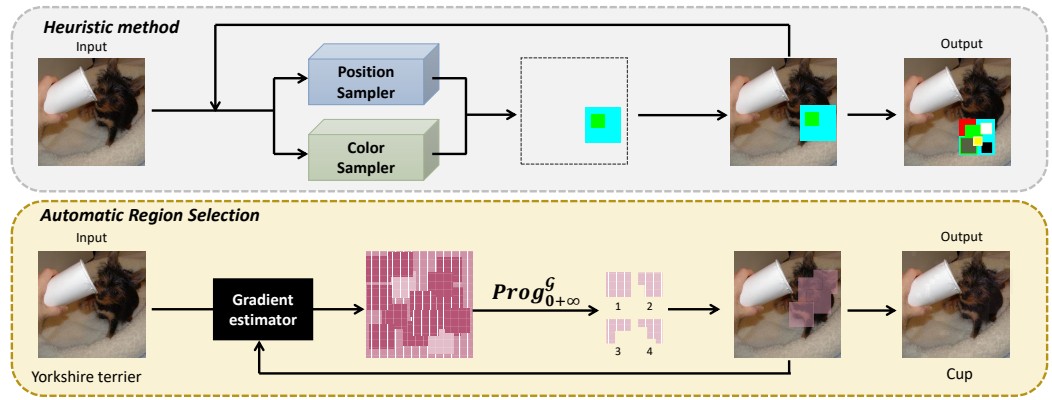

Figure 2: Comparison between automatic region selection attacks and heuristic method in the ImageNet dataset. The top half of the figure displays the perturbation sampled to determine the position and color. The bottom half showcases our method, automatically selecting multiple subregions.

## 2.1 ADVERSARIAL ATTACK

Let $\mathcal{C}(x) : \mathbb{R}^d \to \mathbb{R}^K$ be a well-trained DNN classification model, where $x \in [0,1]^d$ represents the original sample (If $x$ is an image, we have $d = w \times h \times c$, where $w$ denotes the image width, $h$ denotes the image height, $c$ denotes the number of color channels), and $K$ denotes the number of image classes. The goal of adversarial attacking is to find a small perturbation $\delta \in \mathbb{R}^d$ for a given image $x_0$ belonging to class $y_0 \in \{1, 2, \cdots, K\}$ such that the model $\mathcal{C}$ classifies the new image $x_0 + \delta$ into a targeted class $y$ $(y \neq y_0)$. Formally, the objective is to find:

$$\underset{k=1,2,\cdots,K}{\arg\max} \ \mathcal{C}_k(x_0 + \delta) = y \quad \text{s.t. } \|\delta\|_p \leq \varepsilon,\ 0 \leq x_0 + \delta \leq 1$$

We denote by $\varepsilon > 0$ the maximal allowable perturbation under the $\ell_p$ norm. In practice, the $\ell_p$ norm is often replaced by the $\ell_2$ or $\ell_\infty$ norm (Carlini & Wagner, 2017; Ilyas et al., 2018a;b; Zhao et al., 2020).

## 2.2 OBJECTIVE FOR ADVERSARIAL ATTACK WITH AUTOMATICAL REGION DETECTION

In this subsection, we propose a new objective function for adversarial attack with automatic region detection. In order to detect the region automatically, an additional $\ell_0^{\mathcal{G}}$ norm group constraint is added. Then, the adversarial attack problem can then be reformulated as follows:

$$\min_\delta f(x_0 + \delta, y) \quad \text{s.t.} \|\delta\|_0^{\mathcal{G}} \leq k, \|\delta\|_\infty \leq \varepsilon,\ 0 \leq x_0 + \delta \leq 1, \tag{1}$$

where $f(\cdot)$ is margin loss function (Carlini & Wagner, 2017), $k$ is group sparsity of perturbation, $\varepsilon$ is the magnitude of the perturbation. In the targeted attack scenario, $y$ is the targeted class $(y \neq y_0)$, which is the true class $(y = y_0)$ in the untargeted attack scenario. The definition of $\|\delta\|_0^{\mathcal{G}}$, the number of non-zero groups in a vector, is as follows.

**Definition 1.** *Suppose $\mathcal{G} = \{G_1, \ldots, G_M\}$ is a set of $M$ groups that can arbitrarily overlap, $G_i \subseteq [d]$ and $\cup_{i=1}^M G_i = \{1, 2, \ldots, d\}$. We use $\mathbb{B}^M$ to represent the space of $M$-dimensional binary vectors and define $\iota : \mathbb{R}^d \to \mathbb{B}^M$, for any $\delta$ in $\mathbb{R}^d$, $\iota(\delta)_i = 1$ if $\delta_i \neq 0$ and $\iota(\delta)_i = 0$ otherwise. We define the incidence matrix $A^{\mathcal{G}} \in \mathbb{B}^{d \times M} : A_{ij}^{\mathcal{G}} = 1$ if $i \in G_j$ and $A_{ij}^{\mathcal{G}} = 0$ otherwise. The group $\ell_0^{\mathcal{G}}$ norm is defined as*

$$\|\delta\|_0^{\mathcal{G}} := \min_{a \in \mathbb{B}^M} \left\{ \sum_{j=1}^M a_j : A^{\mathcal{G}} a \geq \iota(\delta) \right\}, \tag{2}$$

*where $A^{\mathcal{G}} a \geq \iota(\delta)$ means that $supp(\delta) \subseteq \cup_{a_j=1} G_j$.*

## 3 PROPOSED METHOD

In this section, we introduce our proposed approach to address the problem (1). We outline the key steps involved in our method. Firstly, we employ the natural evolutionary strategy to estimate the gradient. Secondly, we reframe the objective problem by employing the first-order Taylor approximation. Lastly, we present a comprehensive algorithmic description, providing a step-by-step account of our method.

## 3.1 NATURAL EVOLUTIONARY STRATEGY

To develop an effective technique, one intuitive strategy is to employ gradient-based methods for generating adversarial examples while minimizing query requirements. Thus, we use the Natural Evolutionary Strategy (NES) (Wierstra et al., 2014), which is a derivative-free optimization approach centered around a search distribution framework. Specifically, given a current point $x$, we utilize a search distribution $\pi(\theta|x)$ to generate a new point $\theta$ from $x$ based on this distribution. Instead of directly minimizing the loss function $f$, we focus on minimizing the expected value $F$ of the loss function under the search distribution $\pi(\theta|x)$. This expected value is defined as follows:

$$F(x, y) := \mathbb{E}_{\pi(\theta|x)}[f(\theta, y)] = \int f(\theta, y)\pi(\theta|x)d\theta,$$

Next, we can compute the gradient of $F(x, y)$ with respect to $x$ using the following approach (Ilyas et al., 2018a):

$$\nabla_x F(x, y) = \mathbb{E}_{\pi(\theta|x)}[f(\theta, y)\nabla_x \log(\pi(\theta|x))]. \tag{3}$$

Following the methodology employed in (Ilyas et al., 2018a; Wierstra et al., 2014; Ye & Zhang, 2019), we select a point near $x$ by introducing Gaussian noise. Specifically, we employ the central difference sampling method to reduce variance. By evaluating the gradient using these $n$ samples, we obtain a variance-reduced gradient estimation, which can be expressed as follows:

$$g = \frac{1}{n} \sum_{i=1}^{n/2} \frac{f(x + \sigma\tau_i, y) - f(x - \sigma\tau_i, y)}{\sigma}\tau_i,$$

where $\tau \sim \mathcal{N}(0, I)$, $\sigma$ is the variance. It is evident that the gradient estimation $g$ is an unbiased estimate of $\nabla_x F(x, y)$, meaning that $\mathbb{E}[g] = \nabla_x F(x, y)$.

## 3.2 SEQUENTIAL APPROXIMATION AND SOLUTIONS TO EACH SUBPROBLEM

We now introduce an efficient approach to minimize $F$ with $\ell_0^{\mathcal{G}}$ and $\ell_\infty$ constraints, utilizing our gradient estimation to obtain an approximate or closed-form solution for $F$. Let's assume that $F$ is a nonconvex function with smoothness. Given the current point $x = x_0 + \delta$, we have the following relationship:

$$F(x_0 + \delta, y) \leq F(x_0 + \delta^t, y) + \nabla_x F(x_0 + \delta^t, y)^T(\delta - \delta^t) + \frac{L}{2}\|\delta - \delta^t\|_2^2,$$

Obviously, to minimize the right-hand side of the inequality, we can solve the following sequential subproblem for each given $\delta^t$:

$$\min_{\|\delta\|_0^{\mathcal{G}} \leq k, l \leq \delta \leq u} \frac{L}{2}\|\delta - S_L(\delta^t)\|_2^2, \text{ where } S_L(\delta^t) = \delta^t - \frac{1}{L}\nabla F(x_0 + \delta^t, y). \tag{4}$$

To simplify the objective, we combine the second and third constraints into a range $\delta \in [l, u]$, where $l = \max(-\varepsilon, -x_0)$ and $u = \min(\varepsilon, 1-x_0)$ since they are both box constraint. And $\|\cdot\|$ denotes $\|\cdot\|_2$ for simplify in this paper. Then, we discuss how to solve each subproblem (4) in the non-overlapping and overlapping settings, respectively.

**Non-overlapping groups.** For non-overlapping groups, we provide a closed-form solution in the Theorem 1. The details of the proof are given in the Appendix B.1. Note that, the closed-form solution can also be obtained by Algorithm 2.

**Theorem 1.** *Let $\Pi_{[l,u]}(\cdot)$ denote the projection onto $[l, u]^d$. We define $\overrightarrow{\mathbf{DIS}}$ as a group of some independent $\mathbf{DIS}_j$, so we have*

$$\mathbf{DIS}_j = [\Pi_{[l,u]}(S_L(\delta^t))]_j^2 - 2[\Pi_{[l,u]}(S_L(\delta^t))]_j S_L(\delta^t)_j, \quad \overrightarrow{\mathbf{DIS}} := \Pi_{[l,u]}(\delta) \odot (\Pi_{[l,u]}(\delta) - 2\delta) \odot I_G.$$

*$\pi(\cdot)$ denotes the indices that sort $\overrightarrow{\mathbf{DIS}}$ in increasing order as groups. The $I_G \in \mathbb{R}^d$ is a boolean map to indicate the position of a set of perturbations. It is denoted as $I_G(i) = 1$, if $i \in G$, and 0 otherwise. The analytical solution under non-overlapping group sparse constraint can be obtained that ($i \in \{1, 2, \cdots, M\}$)*

$$\delta_{G_i}^{t+1} = \begin{cases} [\Pi_{[l,u]}(S_L(\delta^t))]_{G_i}, & i = \pi(1), \pi(2), \cdots, \pi(k); \\ 0, & otherwise. \end{cases}$$

---

**Algorithm 1** FTAS$\ell_{0+\infty}^{\mathcal{G}}$

---

**Input:** Initial image $x_0$, target class $y_t$, classifier $\mathcal{C}(y|x)$, sparsity $k$, learning rate $\eta$, number of samples $n$, search variance $\sigma$

**Output:** Adversarial image $x_{adv}$ with $\|x_{adv} - x_0\|_0^{\mathcal{G}} \leq k$, $\|x_{adv} - x_0\|_\infty \leq \varepsilon$

1: **Init** $x_{adv}, \delta^0, t$
2: **while** $\max_y \mathcal{C}(y|x_{adv}) \neq y_t$ **do**
3:     **for** $i = 1$ to $n/2$ **do**
4:         $\tau_i \leftarrow \mathcal{N}(\mathbf{0}, I)$
5:         $g_i = \frac{1}{2\sigma}(f(x_{adv} + \sigma\tau_i, y_t) - f(x_{adv} - \sigma\tau_i, y_t))\tau_i$
6:     **end for**
7:     $g = \frac{1}{n}\sum_{i=1}^{n} g_i$
8:     $\tilde{\delta}^{t+1} = \delta^t - \eta g$
9:     $\overrightarrow{\mathbf{DIS}} = \Pi_{[l,u]}(\tilde{\delta}^{t+1}) \odot (\Pi_{[l,u]}(\tilde{\delta}^{t+1}) - 2\tilde{\delta}^{t+1}) \odot I_G$
10:                 ▷ $\odot$ denotes the Hadamard product
11:     $\delta^{t+1} = \Pi_{[l,u]}(\mathcal{P}_k^{\mathcal{G}}(\overrightarrow{\mathbf{DIS}}, \tilde{\delta}^{t+1}))$    ▷ Algorithm 2
12:     $x_{adv} = x_0 + \delta^{t+1}$
13:     $t = t + 1$
14: **end while**

---

**Algorithm 2** $\mathcal{P}_k^{\mathcal{G}}(\overrightarrow{\mathbf{DIS}}, \delta)$

---

**Input:** Group sparsity $k$, Perturbations after truncation $v$, Groups set $\mathcal{G}$, Set of selected groups $\hat{G}$

**Output:** $v$

1: $v = \mathbf{0}, \hat{G}^0 = \varnothing$
2: **for** $i = 1$ to $k$ **do**
3:     $G_{opt}^i = \arg\min_{G \in \mathcal{G} \setminus \hat{G}^{i-1}} Dis_G$
4:     $\overrightarrow{\mathbf{DIS}} = \overrightarrow{\mathbf{DIS}} - \overrightarrow{\mathbf{DIS}}_{G_{opt}^i}$
5:     $v = v + \delta_{G_{opt}^i}$
6:     $\delta = \delta - \delta_{G_{opt}^i}$
7:     $\hat{G}^i = \hat{G}^{i-1} \bigcup G_{opt}^i$
8: **end for**

---

**Overlapping groups.** For overlapping groups, we propose an approximate solution outlined in Algorithm 2. In each iteration of the greedy selection process, we choose a group based on the $\overrightarrow{\mathbf{DIS}}$ value as defined in Theorem 1. For instance, if a pixel is initially selected and belongs to multiple groups, the $\overrightarrow{\mathbf{DIS}}$ value for other groups containing this pixel will be recalculated during the subsequent steps. Additionally, to avoid redundancy, perturbation points that have already been selected are not chosen again in subsequent iterations. The details of the proof are given in the Appendix B.2.

### 3.3 ALGORITHM

In this section, we present our algorithm for solving problem (4), named FTAS$\ell_{0+\infty}^{\mathcal{G}}$ (First-order Taylor Approximation Strategy with $\ell_0^{\mathcal{G}}$ and $\ell_\infty$ constraints). The pseudocode of FTAS$\ell_{0+\infty}^{\mathcal{G}}$ is presented in Algorithm 1. In Line 1 in Algorithm 1, the initial value of $\delta$ is a random variable under a uniform distribution, and then the desired k group of perturbations is selected according to $\overrightarrow{\mathbf{DIS}}$. Each iteration of our algorithm consists of two steps: (i) the NES gradient estimation step (Lines 3-7), and (ii) calculating the solution of each subproblem step, where the NES gradient estimation step is the one described in subsection 3.1. Calculating the solution of each subproblem can also be divided into three steps in implementation: (i) perform gradient updating on $\delta^k$, (ii) calculate $\overrightarrow{\mathbf{DIS}}$ according to Line 9 of Algorithm 1, (iii) get the smallest $k$ groups index according to the value of $\overrightarrow{\mathbf{DIS}}$, reserve the corresponding index $\tilde{\delta}^{t+1}$, and the others are 0, and (iiii) clip the result in $\max\{l, \min\{u, \tilde{\delta}^{t+1}\}\}$ to get the $\Pi_{[l,u]}(\tilde{\delta}^{t+1})$. Details of the implementation of step (iii) are shown in Algorithm 2. This ensures that all iterations of perturbations are under the $k$-groups sparsity and within the $\ell_\infty$ constraint. In Algorithm 2, we select the group with minimum $\overrightarrow{\mathbf{DIS}}$ greedily. For non-overlapping groups, we can obtain the closed-form solution in Theorem 1.

On the other hand, given a vector $\delta \in \mathbb{R}^d$ that requires projection onto the constraint set $\|\delta\|_0^{\mathcal{G}} \leq k$ and $l \leq \delta \leq u$, we encounter an NP-hard problem when $\mathcal{G}$ constrains arbitrary overlapping groups, rendering it challenging to solve the problem (4). We can obtain an approximate solution from Algorithm 2, and provide theoretical guarantees under the standard assumptions when applied to the overlapping group sparsity problem. The convergence analysis is detailed in the Appendix C.

## 4 THEORETICAL PERFORMANCE BOUNDS

In the following, we present the convergence analysis for Algorithm 1. First, we give two important assumptions used in our analysis.

**Assumption 1.** (RSC/RSS). *The function $f : \mathbb{R}^d \to \mathbb{R}$ satisfies the restricted strong convexity (RSC) and restricted strong smoothness (RSS) of order $k^* + k$, which can be expressed as the following : $\alpha_{k^*+k}I \preceq H(\delta) \preceq L_{k^*+k}I$, where $H(\omega)$ is the Hessian of $f$ at any $\delta \in \mathbb{R}^d$ s.t. $\|\delta\|_0^{\mathcal{G}} \leq k^* + k$.*

RSC and RSS conditions have been widely studied in high dimensional statistical theory (Raskutti et al., 2010; Loh & Wainwright, 2013; Agarwal et al., 2010). They guarantee that the objective function behaves like a strongly convex and smooth function over a sparse domain even if the function is non-convex.

**Assumption 2.** $f(x_0 + \delta, y)$ *is bounded on its domain, that is, there exists a generic constant $B > 0$ such that:* $\forall \delta \in \mathbb{R}^d, l \le \delta \le u : |f(x_0 + \delta, y)| \le B$.

Based on the above assumptions, we can now offer theoretical assurances for Algorithm 1:

**Theorem 2.** *Let $\delta^*$ denote the optimal solution to the problem (1), $k^*$ denotes $\|\delta^*\|_0^{\mathcal{G}}$, we can set $\hat{k} = O\left(k^* \log\left(\|\delta^*\| / \xi\right)\right)$ (to ensure that for all $\tilde{\delta}_t$ $\xi \ge e^{-\frac{\hat{k}}{k^*}} \|\tilde{\delta}_t\|_2$), $\eta = \frac{1}{L_{k+k^*}}$, under Assumptions 1 and 2 we have a geometric convergence rate, of the following form:*

$$\mathbb{E}\|\delta_T - \delta^*\| \le \rho^T \mathbb{E}\|\delta_0 - \delta^*\| + \left(\frac{1}{1-\rho}\right) \cdot (a + b + c),$$

*where $\rho = (1 + \sqrt{\frac{k^*}{k-\hat{k}}})(1 - \frac{\alpha_{k+k^*}}{L_{k+k^*}})$, $a = \frac{1+\sqrt{k^*/(k-\hat{k})}}{L_{k+k^*}} \cdot (\sqrt{d}L_{k+k^*}\sigma + \frac{\sqrt{d}B}{\sqrt{n}\sigma})$, $c = \sqrt{\frac{k^*\xi}{k-\hat{k}}}$, $b = \frac{1+\sqrt{k^*/(k-\hat{k})}}{L_{k+k^*}} \cdot \max\{\|\nabla f(\delta^*)_G\|_2 \mid G = \cup_{i=1}^{\tilde{k}} G_{j_i}, G_{j_i} \in \mathcal{G}, \tilde{k} \le k + k^*\}$.*

**Remark 1.** *Let $k = O(\frac{L_{k+k^*}^2}{\alpha_{k+k^*}^2} k^* + \hat{k})$ and set $\sigma$, $n$ appropriately, then the output of Algorithm 1 after $T = O(\frac{L_{k+k^*}^2}{\alpha_{k+k^*}^2} \cdot \log \frac{1}{\xi})$ iterations satisfies*

$$\|\delta_T - \delta^*\| \le 3\xi + \frac{L_{k+k^*}^2}{\alpha_{k+k^*}^2} b.$$

*The approximation errors of the quantity $a$ in Theorem 2 are induced by two factors: the first one is the approximation of the true function of $f(x, y)$ by the function $F(x, y)$, and the second one is the approximation of $\nabla F(x, y)$ via sample average approximations. If $\arg\min_\delta f(x_0 + \delta, y) \in [l, u]^d$ and $\|\arg\min_\delta f(x_0 + \delta, y)\|_0^{\mathcal{G}} \le k$, then $b = 0$. $\xi$ comes from the greedy hard-threshold process in Algorithm 2, in theory $\xi$ can be arbitrarily small if we set $\hat{k} = O\left(k^* \log\left(\|\delta^*\| / \xi\right)\right)$. And if $\mathcal{G}$ is non-overlapped, $c = \xi = 0$. As we reduce $\xi$, the value of $\hat{k}$ increases, so does $k$, which leads to an increase in $b$. So there is a trade-off between the estimation error represented by $\xi$ and the model selection error indicated by $b$. We prove the theorem and the remark in the Appendix C.*

## 5 EXPERIMENTS

In this section, we conduct a comprehensive comparison of our proposed method with black-box adversarial attack methods in the targeted scenario (see Appendix D.6 for untargeted results). Firstly, querying a model will cost expensive money and resources in the real world. Thus, we are interested in query-efficient algorithms for generating adversarial examples. On the CIFAR10 and MNIST datasets, we set the maximum number of queries to 10k for untargeted scenarios, 20k for targeted scenarios, and 20k and 40k for the ImageNet dataset. Secondly, we also pay attention to the imperceptibility of perturbations. We add another $\ell_\infty$ norm to control how visible perturbations are to the human eye. Thus, our proposed method provides better structure and insights into vulnerable regions compared to single-constrained attacks.

### 5.1 BASELINE METHODS

In this section, we conduct a comprehensive evaluation of the performance between our proposed method and various attack modes, including global, region-wise, and pixel-wise sparse attack modes. Specifically, we consider two types of global attacks: gradient estimation methods and heuristic methods. For gradient estimation, we compare with the Zeroth-Order Natural Gradient Descent attack (ZO-NGD)(Zhao et al., 2020), which imposes an $\ell_\infty$ constraint. On the other hand, Parsimonious Attack (Moon et al., 2019) and Square Attack (Andriushchenko et al., 2020) with the boundary of $\ell_\infty$ constraint are used as heuristics to compare our algorithm. Both attacks operate on the entire image region. We provide detailed analysis and results by comparing ASR, $\ell_0$, $\ell_2$, and $\ell_\infty$ norm to demonstrate that we just need fewer perturbations at a similar success rate.

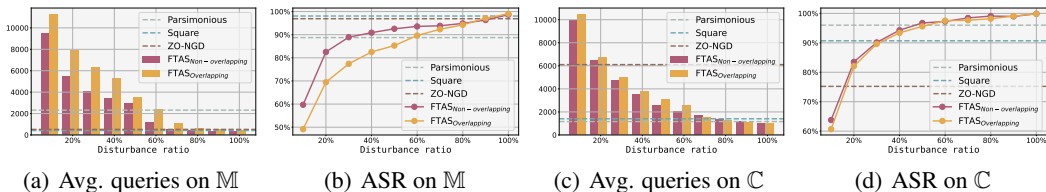

Figure 3: Average query count and Attack Success Rate (ASR) achieved by our algorithm on MNIST ($\mathbb{M}$) and CIFAR10 ($\mathbb{C}$) datasets under varying disturbance ratio, with $\varepsilon = 0.1$ on CIFAR10 datasets, $\varepsilon = 0.4$ on MNIST datasets. The $\ell_\infty$ constraints of other algorithms are the same as ours.

In addition to the global attacks, we also evaluate our proposed method under region-fixed and region-wise attack modes. Specifically, we conduct Fixed-ZO-NGD, Fixed-Square, and Fixed-Parsimonious, which focus on a fixed region of the image. We evaluate ASR, average, and median queries to demonstrate that our method has a higher success rate with fewer queries than state-of-the-art attack algorithms under fixed regions. Furthermore, we compare with the Patch-RS in Sparse-RS (Croce et al., 2022) within the same perturb pixels, which heuristically found the location of the patch. In addition, we utilize visual presentation to demonstrate the structure of our perturbation set compared with pixel-wise sparse attack methods SZOHT (de Vazelhes et al., 2022).

Each of these attack types has its advantages and disadvantages, as summarized in Tab. **??**. Our evaluation provides insights into the effectiveness and better structure of our proposed method over other different attack modes. Due to space limitations, we defer the results on the ImageNet dataset and ablation experiments to Appendix D.7 and Appendix E.

Table 1: Characteristics of different types of attacks.

| Attack Type | Description | Visibility | Objective |
| --- | --- | --- | --- |
| Global | Alters entire image uniformly. | Highly | Degrade overall image quality. |
| Regional | Targets specific areas. | Moderately | Conceal or alter specific parts. |
| Sparse Pixel-wise | Alters a few scattered pixels. | Least | Sparse and conspicuous disturbance. |

## 5.2  RESULT AND ANALYSIS

As demonstrated in the previous section, we conduct a comprehensive comparison of global attack methods including Parsimonious, Square$\ell_\infty$, and ZO-NGD attacks. As shown in Tab. 2 and Fig. 3. For region-wise attacks, we compare with region-fixed global algorithms and Patch-RS of Sparse-RS. It was shown in Tab. 3 and Fig. 4. For pixel-wise sparse attack mode, we give a visual presentation in Fig. 5.

**Global attack mode:** As shown in Tab. 2, we can obtain that our method exhibits a more significant performance than other algorithms when the proportion of the disturbed image reaches 100%, that is the global perturbation. And achieve a similar ASR to Square$\ell_\infty$ and better than ZO-NGD at a 30% ratio of perturbation. From Fig. 3, we can see that we need to strike a balance on the constraint boundary to get low queries and high ASR. Before we reach a 100% perturbing ratio, we can outperform other algorithms in ASR and Avg. query performance. We generate sparse and imperceptible perturbations through controllable constraints, and under tight query budgets, we can achieve higher ASR by perturbing fewer pixels than global perturbations.

**Region-wise attack mode:** In Tab. 3, Patch-RS achieved a great performance by heuristically drawing a square patch on the image, but this patch is easily detectable to the human eye. Our overlapping group algorithm performs better than others. From Fig. 4, we can see that when $\varepsilon$ reaches $0.5 \sim 0.6$, the performance of our overlapping group both in Avg. query and ASR exceeds that of Patch-RS. As can be seen from the $\ell_2$ and $\ell_\infty$ distance, the imperceptibility of our method is stronger.

In the region-fixed attack mode, we maintain the same constraint, i.e., the same amount of perturbation and the same magnitude of perturbation. As shown in Tab. 4, our algorithm is much better than other region-fixed algorithms under the same strict constraints and query budget. For both the MNIST and CIFAR10 datasets, the object of interest typically occupies a significant portion of the

Table 2: Comprehensive comparison of global attack algorithms with $\ell_\infty$ constraints on CIFAR10 and MNIST, where $\varepsilon = 0.4$ in MNIST, $\varepsilon = 0.1$ in CIFAR10.

| | CIFAR10 | | | | | MNIST | | | | |
|---|---|---|---|---|---|---|---|---|---|---|
| **Algorithm** | **ASR** | **Avg.** | **Med.** | $\ell_0$ | $\ell_2$ | **ASR** | **Avg.** | **Med.** | $\ell_0$ | $\ell_2$ |
| Parsimonious | 96.00% | 1164.3 | 212.0 | 3061.0 | 5.4 | 88.74% | 2318.4 | **65.0** | 227.7 | 5.5 |
| Square$\ell_\infty$ | 90.69% | 1405.5 | **103.0** | 3053.9 | 5.4 | 98.07% | 419.5 | 103.0 | 478.6 | 8.3 |
| ZO-NGD | 75.20% | 6105.2 | 707.0 | 3055.3 | 5.4 | 96.90% | 522.9 | 101.0 | 469.6 | 8.2 |
| Ours(N)$_{100\%d}$ | **99.40%** | **963.4** | 387.5 | 3051.4 | 5.1 | 98.66% | **286.4** | 93.5 | 511.9 | 8.0 |
| Ours(O)$_{100\%d}$ | 98.67% | 980.9 | 379.0 | 3057.7 | 5.0 | **99.03%** | 412.9 | 104.5 | 541.0 | 7.8 |
| Ours(N)$_{30\%d}$ | 90.13% | 4684.3 | 1353.5 | **923.4** | **2.9** | 87.28% | 4531.8 | 1076.0 | **197.9** | **5.3** |
| Ours(O)$_{30\%d}$ | 91.61% | 4999.0 | 1470.5 | **908.4** | **2.8** | 77.40% | 6290.3 | 1581.0 | **176.4** | **4.9** |

[*] (N) Non-overlapping groups; (O) Overlapping groups; Number%d: the proportion of perturbed image features.

Table 3: Comprehensive comparison of Sparse-RS (Patch-RS) algorithms with $\ell_0$ constraints on MNIST and CIFAR10. The perturbation ratio of the image is 10% of all features for all algorithms.

| | CIFAR10 | | | | | MNIST | | | | |
|---|---|---|---|---|---|---|---|---|---|---|
| **Algorithm** | **ASR** | **Avg.** | **Med.** | $\ell_2$ | $\ell_\infty$ | **ASR** | **Avg.** | **Med.** | $\ell_2$ | $\ell_\infty$ |
| Patch-RS | 92.51% | 1954.0 | **74.0** | 8.9 | 0.9 | 88.45% | 2953.8 | 146.0 | 6.3 | 0.9 |
| Ours(N)$_{\epsilon=1}$ | **93.16%** | 2386.7 | 1714.0 | 8.4 | 0.9 | 99.25% | 931.2 | 373.5 | 7.9 | 1.0 |
| Ours(O)$_{\epsilon=1}$ | **99.08%** | **1537.7** | 352.0 | 12.6 | 1.0 | **100.00%** | **359.3** | **142.5** | 10.7 | 1.0 |
| Ours(N) | 72.60% | 8960.2 | 4822.0 | **2.8** | **0.2** | 82.00% | 5450.5 | 1266.5 | **4.1** | **0.5** |
| Ours(O) | 84.58% | 6638.4 | 3369.0 | **3.5** | **0.2** | 91.51% | 3513.0 | 1009.0 | **4.9** | **0.5** |

[*] (N) Non-overlapping groups; (O) Overlapping groups; Our without footnotes indicates $\varepsilon = 0.2$ for CIFAR10, $\varepsilon = 0.5$ for MNIST.

image. Consequently, selecting the attack region fixed at the center of the image yields better results compared to other locations. To showcase the exceptional efficiency of our algorithm comprehensively, we provide additional results for perturbations in other locations in the Appendix E.1.

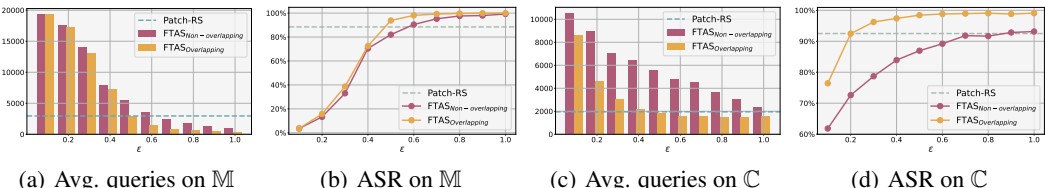

| (a) Avg. queries on $\mathbb{M}$ | (b) ASR on $\mathbb{M}$ | (c) Avg. queries on $\mathbb{C}$ | (d) ASR on $\mathbb{C}$ |

Figure 4: Average query count and Attack Success Rate (ASR) achieved by our algorithm on MNIST ($\mathbb{M}$) and CIFAR10 ($\mathbb{C}$) datasets under different disturbance amplitude. Only 10% of dimensions have been perturbed on both datasets. The total number of maximum perturbed pixels is the same for all algorithms.

**Pixel-wise attack mode:** In the last column of Fig. 5, we present the visual renderings of the adversarial examples generated by SZOHT. It is evident that SZOHT introduces sparse perturbations across global pixels. But intuitively, these perturbations may not exhibit a direct connection to the target class.

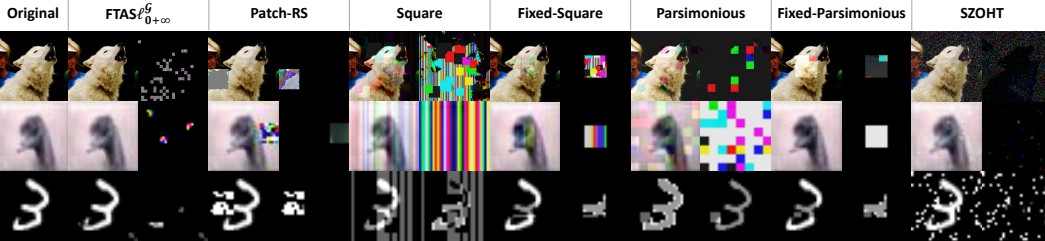

Figure 5: Adversarial examples and the corresponding perturbation on ImageNet datasets crafted by all baseline methods when attacking the Inception-v3 model in the black-box setting with a random-selected target.

Table 4: Comprehensive comparison of region-fixed targeted attack algorithms with $\ell_{0+\infty}$ constraints on MNIST and CIFAR10, where $\varepsilon = 0.4$ in MNIST, $\varepsilon = 0.1$ in CIFAR10. The perturbation ratio of the image is 10% of all features for all algorithms.

| Algorithm | CIFAR10 | | | MNIST | | |
|---|---|---|---|---|---|---|
| | ASR | Avg. | Med. | ASR | Avg. | Med. |
| Fixed-Parsimonious | 26.00% | 15062.5 | 20000.0 | 6.79% | 18717.1 | 20000.0 |
| Fixed-Square$\ell_\infty$ | 57.56% | 11599.6 | 8754.0 | 17.38% | 16630.1 | 20000.0 |
| Fixed-ZO-NGD | 33.50% | 10688.3 | 20000.0 | 36.12% | 14573.9 | 20000.0 |
| Ours(N)$_{10\%d}$ | **63.77%** | **9944.6** | **7512.0** | **59.70%** | **9476.7** | **4768.0** |
| Ours(O)$_{10\%d}$ | **60.65%** | 10488.5 | 7892.0 | 50.13% | 11289.9 | 19936.5 |

[*] (N) Non-overlapping groups; (O) Overlapping groups; Number%d: the proportion of perturbed image features.

**Region-wise attack on ImageNet:**

In this study, we present the performance of region-wise adversarial attacks on the ImageNet dataset, which are concisely summarized in the subsequent table. A detailed analysis can be obtained in Part D in the appendix. It was observed that attacks targeting high-resolution images exhibit lower success rates and necessitate a greater number of queries, particularly when subject to double constraints. The complexity of high-resolution images and intricate network architectures underscore the need for more refined optimization strategies in adversarial attacks. Notably, in scenarios involving non-overlapping groups, our proposed methodology demonstrates a distinct advantage by offering a closed-form solution, in contrast to the heuristic approach.

Table 5: Performance of region-based attack patterns on the ImageNet dataset

| Algorithm | Inception-v3 | | | ViT-B/16 | | |
|---|---|---|---|---|---|---|
| | ASR | Avg. | Med. | ASR | Avg. | Med. |
| Patch-RS | 92.29% | 2968.6 | 685.5 | 86.90% | 3572.8 | 1849.5 |
| Ours(N)$_{\epsilon=1}$ | **98.95%** | 2756.8 | **312.0** | 98.89% | **2682.5** | 1478.0 |
| Ours(O)$_{\epsilon=1}$ | 98.02% | **1927.4** | 406.5 | **99.84%** | 3896.7 | **1008.5** |
| Fixed-Parsimonious | 74.46% | 9631.3 | 7543.0 | 85.36% | 8248.5 | 4952.5 |
| Fixed-Square$\ell_\infty$ | 75.53% | 8893.3 | 2984.5 | 78.95% | 8624.8 | **2286.0** |
| Fixed-ZO-NGD | 79.12% | 7436.0 | **1500.0** | 80.98% | 9426.0 | 3789.5 |
| Ours(N)$_{10\%d,\epsilon=0.1}$ | 76.89% | 7202.8 | 5117.7 | 77.26% | 6796.3 | 5470.0 |
| Ours(O)$_{10\%d,\epsilon=0.1}$ | **83.15%** | **5298.4** | 2965.0 | **89.40%** | **6300.5** | 3238.0 |

## 6 CONCLUSION

In conclusion, we presented a novel approach for region-wise adversarial attacks in the black-box setting. By utilizing automatic region selection, and controllable imperceptibility, our proposed method showed improved effectiveness and interpretability compared to existing attack modes. Experimental evaluations demonstrated that the method required fewer perturbations and queries while achieving higher success rates. We provide valuable insights into understanding vulnerable regions and enhancing the robustness of deep neural networks against adversarial attacks. Of course, we acknowledge that different groupings will have a great impact on the results, and in the future, we will explore the combination of this method with techniques such as image segmentation and principal component analysis to explore the robustness and fragility of neural networks from more perspectives.

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

APPENDIX FOR "QUERY EFFICIENT BLACK-BOX ADVERSARIAL ATTACK WITH AUTOMATIC REGION SELECTION"

The appendix contains several additional results that were excluded from the main body of the paper due to space constraints, along with the proof process of the lemma and solutions. The organization of the appendix is as follows:

**Part A Related Work:** This section provides a discussion and review of relevant existing work in the field.

**Part B Proof for Closed-form and Approximate Solution:** Here, the appendix outlines the proof for the closed-form and approximate solutions proposed in the paper.

**Part C Proof of Convergence:** This section provides a detailed proof procedure for convergence analysis.

**Part D Supplementary experiment:** This section provides details of the setup of the experiment, the performance of untargeted attacks on CIFAR10 and MNIST datasets, and the performance on the ImageNet dataset.

**Part E Ablation experiment:** This section provides a comprehensive evaluation of region-fixed algorithms that are fixed to different regions. In addition, there was an ablation assessment of the size of the group window.

**Part F Supplementary:** This section provides confirmatory experiments on approximate solutions, as well as histograms of frequency distributions for query distributions.

## A    RELATED WORK

In the previous adversarial work, designing imperceptible perturbations was the main goal. With the deepening of our work, we find that exploring the vulnerable regions of the image is an important part of the work. The exploration of the vulnerable region and explanatory properties of disturbances remains at the empirical level. Indeed, exploring the interpretability of adversarial attacks and the vulnerability of neural networks is indeed a novel and important research direction.

In recent years, some structural attacks have emerged in white-box attacks, which capture and leverage structural information hidden in the input (Kazemi et al., 2023; Xu et al., 2018; Zhu et al., 2021). For instance, FWnucl (Kazemi et al., 2023) introduced nuclear norm regularization to promote structured sparsity in perturbations. By incorporating structure-enhancing algorithms, they investigated distortion sets that exhibit enhanced structural properties. This work highlights the potential benefits of incorporating structural information into adversarial attack methods. Homotopy-Attack (Zhu et al., 2021) extends its algorithm to include group-wise sparsity and adapts the nmAPG solution.

Furthermore, StrAttack (Xu et al., 2018) introduced a structural loss term to generate perturbations that preserve important image features. This was achieved through the partitioning of groups based on coordinates, where perturbations within the same group are spatially continuous. By enforcing this structural constraint, StrAttack produces perturbations that are visually meaningful and interpretable. However, It cannot control the number of perturbed regions exactly since it uses the $\ell_1$ norm on the group norms while we use the $\ell_0$ norm. That means it will generate more perturbations on the images and cause greater damage to images. Although it can use different thresholds to adjust how many groups to select, it needs to carefully tune the hyperparameters to get the appropriate hyperparameter. This leads to a very high time complexity compared to our method.

To the best of our knowledge, there has been relatively limited research on black-box attacks specifically aimed at exploring vulnerable regions within images. From a technical standpoint, black box attacks present significant challenges as they inherently lack effective solutions, making the generation of interpretable perturbations a formidable task. This challenge arises from the complexities involved in both optimization techniques and attack strategies.

In addition, there is also some work that continues to explore the performance of the physical world of black-box adversarial attacks. They focus more on the meaning of methods in the physical world, mainly in the areas of face recognition or traffic sign recognition. For instance, (Feng et al., 2022)

proposed GRAPHITE that can automatically generate small masks and optimize with gradient-free optimization. (Wei et al., 2022) proposed a method to simultaneously optimize the position and perturbation for an adversarial patch in the black-box scenario.

# B  SOLUTION

## B.1  PROOF FOR NON-OVERLAPPING GROUPS

*Proof.* We divide the proof into two steps below.

***First step.*** Let's first consider the following optimization problem for the non-overlapping fixed set $\mathcal{A}$, where $\mathcal{A} \subseteq \mathcal{G}, |\mathcal{A}| \leq k$. We define that $l_{G_i} \leq \delta_{G_i} \leq u_{G_i}$ if $\forall G_i \in \mathcal{A}$, and $\delta_{G_i} = 0$ otherwise. Then the problem can be written as below:

$$\min_{\delta} \frac{L}{2} \|\delta - S_L(\delta^t)\|^2$$
$$s.t. \ \delta_{G_i} = 0, \forall G_i \notin \mathcal{A}; \quad l_{G_i} \leq \delta_{G_i} \leq u_{G_i}, \forall G_i \in \mathcal{A} \tag{5}$$

Let $G_i$ be arbitrarily chosen. One can observe that the objective function and the constrained set of problems are both separable. Using this fact, the problem (5) can be transformed as

$$\min_{l \leq \delta \leq u} \frac{L}{2} \sum_{G_i \in \mathcal{G}} \sum_{j \in G_i} (\delta_j - S_L(\delta^t)_j)^2$$

$$= \min_{l \leq \delta \leq u} \frac{L}{2} \sum_{G_i \in \mathcal{A}} \sum_{j \in G_i} (\delta_j - S_L(\delta^t)_j)^2 + \frac{L}{2} \sum_{G_i \in \mathcal{G} \backslash \mathcal{A}} \sum_{j \in G_i} (\delta_j - S_L(\delta^t)_j)^2$$

$$= \min_{l \leq \delta \leq u} \frac{L}{2} \sum_{G_i \in \mathcal{A}} \sum_{j \in G_i} (\delta_j - S_L(\delta^t)_j)^2 + \frac{L}{2} \sum_{G_i \in \mathcal{G} \backslash \mathcal{A}} \sum_{j \in G_i} S_L(\delta^t)_j^2$$

It is obvious that the closed solution to the above-constrained optimization problem is ($i \in \{1, 2, 3, \cdots, M\}$)

$$\delta_{G_i}^* = \begin{cases} [\Pi_{[l,u]}(S_L(\delta^t))]_{G_i}, & G_i \in \mathcal{A}; \\ 0, & G_i \in \mathcal{G} \backslash \mathcal{A}. \end{cases}$$

***Second step.*** The next task, we need to find the optimal set $\mathcal{A}$. The optimization problem can be transformed as

$$\min_{\mathcal{A}} \frac{L}{2} [\sum_{G_i \in \mathcal{A}} \sum_{j \in G_i} ([\Pi_{[l,u]}(S_L(\delta^t))]_j - S_L(\delta^t)_j)^2 + \sum_{G_i \in \mathcal{G} \backslash \mathcal{A}} \sum_{j \in G_i} S_L(\delta^t)_j^2]$$

$$= \min_{\mathcal{A}} \frac{L}{2} \sum_{G_i \in \mathcal{A}} \sum_{j \in G_i} [\Pi_{[l,u]}(S_L(\delta^t))]_j^2 - 2[\Pi_{[l,u]}(S_L(\delta^t))]_j S_L(\delta^t)_j + \frac{L}{2} \sum_{G_i \in \mathcal{G}} \sum_{j \in G_i} S_L(\delta^t)_j^2$$

We define

$$\mathbf{DIS}_j = [\Pi_{[l,u]}(S_L(\delta^t))]_j^2 - 2[\Pi_{[l,u]}(S_L(\delta^t))]_j S_L(\delta^t)_j,$$

$$\mathbf{DIS}_{G_i} = \sum_{j \in G_i} \mathbf{DIS}_j,$$

where $i \in \{1, 2, \cdots, M\}$. Since $\forall j \in G_i, l_j \leq 0$ and $u_j \geq 0$, we have

- if $S_L(\delta^t)_j \in [l_j, u_j]$, then
$$\mathbf{DIS}_j = -S_L(\delta^t)_j^2 \leq 0;$$

- if $S_L(\delta^t)_j < l_j \leq 0$, then
$$\mathbf{DIS}_j = l_j^2 - 2l_j S_L(\delta^t)_j$$
$$= l_j^2 - 2l_j(l_j + S_L(\delta^t)_j - l_j)$$
$$= -l_j^2 - 2l_j(S_L(\delta^t)_j - l_j) \leq 0;$$

- if $S_L(\delta^t)_j > u_j \geq 0$, then

$$
\begin{aligned}
\mathbf{DIS}_j &= u_j^2 - 2u_j S_L(\delta^t)_j \\
&= u_j^2 - 2u_j(u_j + S_L(\delta^t)_j - u_j) \\
&= -u_j^2 - 2u_j(S_L(\delta^t)_j - u_j) \leq 0.
\end{aligned}
$$

So in all cases, we have $\mathbf{DIS}_j \leq 0, \mathbf{DIS}_{G_i} \leq 0$. Then we sort the values of $\mathbf{DIS}_{G_i}$ in increasing order $\pi(\cdot)$.

$$
\mathbf{DIS}_{\pi(1)} \leq \mathbf{DIS}_{\pi(2)} \leq \cdots \leq \mathbf{DIS}_{\pi(k)} \leq \cdots \leq \mathbf{DIS}_{\pi(M)} \leq 0.
$$

Thus, $\mathcal{A} = \{G_{\pi(1)}, G_{\pi(2)}, \cdots, G_{\pi(k)}\}$ can be obtained by truncating indexes of the smallest $k$ entires of $\mathbf{DIS}_{\mathcal{G}_\pi}$, where $\mathcal{G}_\pi = \{G_{\pi(1)}, G_{\pi(2)}, \cdots, G_{\pi(k)}, \cdots, G_{\pi(M)}\}$ ($k \leq M$).

***Conclusion.*** Through the above proof, the analytical solution under non-overlapping group sparse constraint can be obtained that ($i \in \{1, 2, 3, \cdots, M\}$)

$$
\delta^*_{G_i} = \begin{cases} [\Pi_{[l,u]}(S_L(\delta^t))]_{G_i}, & i = \pi(1), \pi(2), \cdots, \pi(k); \\ 0, & otherwise. \end{cases}
$$

$\square$

### B.2 Proof for Overlapping Groups

From problem (4), let's first consider the following optimization problem for the overlapping fixed set $\mathcal{A}$, where $\mathcal{A} \subseteq \mathcal{G}, |\mathcal{A}| \leq k$. We define that $l_j \leq \delta_j \leq u_j$ if $\forall j \in \bigcup_{G_i \in \mathcal{A}} G_i$, and $\delta_j = 0$ otherwise. Then the problem can be written as below:

$$
\begin{aligned}
&\min_\delta \frac{L}{2} \|\delta - S_L(\delta^t)\|^2 \\
&s.t. \ \delta_j = 0, \forall j \notin \cup_{G_i \in \mathcal{A}} G_i; \ l_j \leq \delta_j \leq u_j, \forall j \in \cup_{G_i \in \mathcal{A}} G_i
\end{aligned} \tag{6}
$$

Let $G_i$ be arbitrarily chosen. One can observe that the objective function and the constrained set of problems are both separable. Using this fact, the problem (6) can be transformed as

$$
\begin{aligned}
&\min_{l \leq \delta \leq u} \frac{L}{2} \sum_{j \in \bigcup_{G_i \in \mathcal{A}} G_i} (\delta_j - S_L(\delta^t)_j)^2 + \frac{L}{2} \sum_{j \in [d] \setminus \bigcup_{G_i \in \mathcal{A}} G_i} (\delta_j - S_L(\delta^t)_j)^2 \\
&= \min_{l \leq \delta \leq u} \frac{L}{2} \sum_{j \in \bigcup_{G_i \in \mathcal{A}} G_i} (\delta_j - S_L(\delta^t)_j)^2 + \frac{L}{2} \sum_{j \in [d] \setminus \bigcup_{G_i \in \mathcal{A}} G_i} S_L(\delta^t)_j^2
\end{aligned} \tag{7}
$$

Obviously, the objective function and the constrained set of problems are both separable by index. For the box constraint $l \leq \delta \leq u$, we can obtain the optimal solution $\Pi_{[l,u]}(S_L(\delta^t))$. Then we rewrite (7) as

$$
\begin{aligned}
&\min_{\mathcal{A}} \frac{L}{2} \Big[ \sum_{j \in \bigcup_{G_i \in \mathcal{A}} G_i} ([\Pi_{[l,u]}(S_L(\delta^t))]_j - S_L(\delta^t)_j)^2 + \sum_{j \in [d] \setminus \bigcup_{G_i \in \mathcal{A}} G_i} S_L(\delta^t)_j^2 \Big] \\
&= \min_{\mathcal{A}} \frac{L}{2} \sum_{j \in \bigcup_{G_i \in \mathcal{A}} G_i} [\Pi_{[l,u]}(S_L(\delta^t))]_j^2 - 2[\Pi_{[l,u]}(S_L(\delta^t))]_j S_L(\delta^t)_j + \frac{L}{2} \sum_{j \in [d]} S_L(\delta^t)_j^2.
\end{aligned}
$$

This is equivalent to

$$
\min_{\mathcal{A}} \sum_{j \in \bigcup_{G_i \in \mathcal{A}} G_i} [\Pi_{[l,u]}(S_L(\delta^t))]_j^2 - 2[\Pi_{[l,u]}(S_L(\delta^t))]_j S_L(\delta^t)_j.
$$

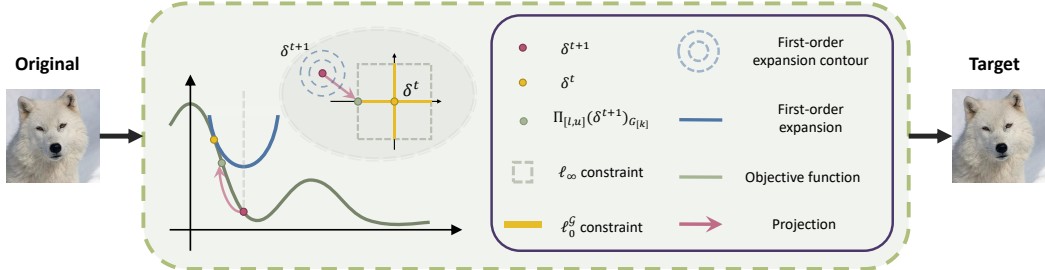

Figure 6: Algorithm Schematic diagram. The original problem is non-convex. The figure illustrates the iterative update process for finding the optimal $\delta^*$ using NES gradient estimation. The search process involves utilizing a first-order Taylor expansion followed by projection onto the imposed constraints. After projection, iterative updates are performed until either a successful attack is achieved or the maximum query limit is reached.

Denote

$$\mathbf{DIS} := \Pi_{[l,u]}(S_L(\delta^t)) \odot \left( \Pi_{[l,u]}(S_L(\delta^t)) - 2S_L(\delta^t) \right), \tag{8}$$

where $\odot$ denotes the Hadamard product. Then we can define a Boolean map $I_G \in \mathbb{R}^d$ to indicate the position of a set of perturbations. It is denoted as $I_G(i) = 1$, if $i \in G$, and 0 otherwise. We define

$$\mathbf{DIS}_G = \mathbf{DIS} \odot I_G. \tag{9}$$

To project $k$ overlapping groups from $\mathcal{G}$, we implement an iterative greedy method similar to IHT, which projects one group at a time and updates the index set to avoid duplicate selection for the same indexes. Thus, in iteration $i(1 \leq i \leq k)$, we can find optimal group $G^i_{opt}$ using

$$G^i_{opt} = \underset{G \in \mathcal{G} \setminus \hat{G}^{i-1}}{\arg \min} \mathbf{DIS}_G, \tag{10}$$

where $\hat{G}^i = \left\{ G^1_{opt}, \ldots, G^i_{opt} \right\}$ denotes already selected groups after $i^{th}$ iteration, with initial value $\varnothing$ and $\hat{G}^k = \mathcal{A}$. The details can be found in Algorithm 2 with the geometric interpretation illustrated in Fig. 6.

### B.3 COST ANALYSIS

In this section, we will analyze the complexity of region selection and optimize the attack with the current region in the case of overlapping groups. In our method, Algorithm 2 is used to determine the region to be attacked which has the time complexity of $P(\hat{k}d)$. Algorithm 1 can be viewed as optimizing the attack with the current region. The total complexity is $O(Tnd)$. The following table lists the detailed time complexity of each row in our algorithm according to Algorithm 1 and Algorithm 2. In the table, $T$ denotes the number of iterations, $d$ denotes the dimension of data, $n$ denotes the population of each estimated gradient, and $\hat{k}$ indicates the number of reserved groups in the constraint.

Table 6: The complexity of optimizing the attack with the current region.

| Lines | Time Complexity |
|---|---|
| 3-6 | $O(nd)$ |
| 7 | $O(n)$ |
| 8-9 | $O(d)$ |
| 12 | Region Selection |
| 16 | $O(d)$ |
| 17 | $O(1)$ |
| **Total** | $O(Tnd)$ |

Table 7: The complexity in the region selection process.

| Lines | Time Complexity |
|---|---|
| 3 | $O(d)$ |
| 4 | $O(d)$ |
| 5 | $O(d)$ |
| 6 | $O(d)$ |
| 7 | $O(d)$ |
| **Total** | $O(kd)$ |

## C    PROOF OF CONVERGENCE

Algorithm 2 can be understood as a two-step projection in a simple way.

**Step-1.** We first select $\hat{k}$ groups according to the following rules:

$$G^i = \underset{G \in \mathcal{G} \setminus \cup_{n=0}^{i-1} G^n}{\arg\min} Dis_G,$$

where $G \in \mathcal{G}$. We specify that $G^0 = \varnothing$, and denote the greedy projection result $\delta_{\cup_{i=1}^{\hat{k}} G^i}$ as $\mathcal{P}_{\hat{k}}^{\mathcal{G}}(\delta)$.

**Step-2.** Projecting $\mathcal{P}_{\hat{k}}^{\mathcal{G}}(\delta)$ onto $[l,u]^d$, using $\Pi_{[l,u]}(\cdot)$ to represent the corresponding projection operator.

### C.1    NOTATIONS AND DEFINITIONS

For the sake of convenience, we give a review of the definitions provided before and defined some new symbols.

- $f(\delta)$ denotes $f(x_0 + \delta, y)$.
- $F(\cdot)$ denotes the Gaussian smoothing of $f(\cdot)$.
- $\nabla F(\delta_t)$ denotes the gradient of $F(\cdot)$ at $\delta_t$.
- $g_t$ denotes the variance-reduced gradient of $F(\cdot)$ at $\delta_t$.
- $\tilde{\delta}_{t+1} = \delta_t - \eta g_t$.
- $p_{t+1}$ denotes $\mathcal{P}_{\hat{k}}^{\mathcal{G}}(\tilde{\delta}_{t+1})$.
- $\Pi(\cdot) : \mathbb{R}^d \to [l,u]^d$.
- $\delta_{t+1} = \Pi(p_{t+1})$.
- $\delta^*$ denotes the optimal solution to the problem (1).
- $k^*$ denotes $\|\delta^*\|_0^{\mathcal{G}}$.
- $\overrightarrow{\mathbf{DIS}}(\delta, G) := \Pi_{[l,u]}(\delta) \odot (\Pi_{[l,u]}(\delta) - 2\delta) \odot I_G$.
- $\mathbf{DIS}^*(\delta, G) = \|\overrightarrow{\mathbf{DIS}}(\delta, G)\|_1$.

### C.2    LEMMAS

**Lemma 1.** *For any vector $\boldsymbol{g} \in \mathbb{R}^d$, using Algorithm 2 to obtain its projection onto the space where $\|\cdot\|_0^{\mathcal{G}} \leq k$, and suppose that $X = \{G_{i_1}, G_{i_2}, \ldots, G_{i_{\hat{k}}}\}$ is the set selected in order according to Algorithm 2, and a set $E \subseteq [d]$ such that there exists $T = \{G_{t_1}, G_{t_2}, \ldots, G_{t_{k^*}}\}$ satisfying $T \subseteq \mathcal{G}$ and $E \subseteq \cup_{j=1}^{k^*} G_{t_j}$, the following inequality must hold*

$$\mathbf{DIS}^*(\boldsymbol{g}, X) \geq (1 - e^{-\frac{\hat{k}}{k^*}})\mathbf{DIS}^*(\boldsymbol{g}, E).$$

*Then, it is obvious that $\mathbf{DIS}^*(\boldsymbol{g}, X) \geq \mathbf{DIS}^*(\boldsymbol{g}, E)$ when $\mathcal{G}$ is non-overlapped and $\hat{k} \geq k^*$.*

*Proof.* It is obvious that the function $\mathbf{DIS}^*(\boldsymbol{g}, \cdot) : 2^{[d]} \to \mathbb{R}$ satisfies the inequality

$$\mathbf{DIS}^*(\boldsymbol{g}, Y \setminus S) \geq \mathbf{DIS}^*(\boldsymbol{g}, Y \setminus T),$$
$$\mathbf{DIS}^*(\boldsymbol{g}, S \cup Y) - \mathbf{DIS}^*(\boldsymbol{g}, S) \geq \mathbf{DIS}^*(\boldsymbol{g}, T \cup Y) - \mathbf{DIS}^*(\boldsymbol{g}, T),$$

where $\forall S, T, Y \in 2^{[d]}, S \subseteq T$. Let $T_m = \cup_{j=0}^m G_{t_j}$, $X_n = \cup_{j=1}^n G_{i_j}$, and $T_0$ and $X_0$ as empty sets. Then, based on the properties mentioned above, and the way $X$ is selected ($G_{i_k} =$

$\arg\min_{G \in \mathcal{G} \setminus \cup_{n=1}^{k-1} G_{i_n}} \mathbf{DIS}_G)$, we have

$$
\begin{aligned}
\mathbf{DIS}^*(\boldsymbol{g}, T_{k^*}) &\leq \mathbf{DIS}^*(\boldsymbol{g}, X_n \cup T_{k^*}) \\
&= \mathbf{DIS}^*(\boldsymbol{g}, X_n \cup T_{k^*}) - \mathbf{DIS}^*(\boldsymbol{g}, X_n) + \mathbf{DIS}^*(\boldsymbol{g}, X_n) \\
&= \sum_{j=1}^{k^*} (\mathbf{DIS}^*(\boldsymbol{g}, X_n \cup T_j) - \mathbf{DIS}^*(\boldsymbol{g}, X_n \cup T_{j-1})) + \mathbf{DIS}^*(\boldsymbol{g}, X_n) \\
&\leq \sum_{j=1}^{k^*} [\mathbf{DIS}^*(\boldsymbol{g}, X_n \cup G_{t_j}) - \mathbf{DIS}^*(\boldsymbol{g}, X_n)] + \mathbf{DIS}^*(\boldsymbol{g}, X_n) \\
&\leq k^* [\mathbf{DIS}^*(\boldsymbol{g}, X_{n+1}) - \mathbf{DIS}^*(\boldsymbol{g}, X_n)] + \mathbf{DIS}^*(\boldsymbol{g}, X_n).
\end{aligned}
$$

Based on this recursive relation, we can obtain that

$$
\begin{aligned}
\mathbf{DIS}^*(\boldsymbol{g}, X) = \mathbf{DIS}^*(\boldsymbol{g}, X_{\hat{k}}) \\
&\geq [1 - (1 - \frac{1}{k^*})^{\hat{k}}] \mathbf{DIS}^*(\boldsymbol{g}, T_{k^*}) \\
&\geq [1 - ((1 - \frac{1}{k^*})^{k^*})^{\frac{\hat{k}}{k^*}}] \mathbf{DIS}^*(\boldsymbol{g}, T_{k^*}) \\
&\geq (1 - e^{-\frac{\hat{k}}{k^*}}) \mathbf{DIS}^*(\boldsymbol{g}, T_{k^*}) \\
&= (1 - e^{-\frac{\hat{k}}{k^*}}) \mathbf{DIS}^*(\boldsymbol{g}, T) \\
&\geq (1 - e^{-\frac{\hat{k}}{k^*}}) \mathbf{DIS}^*(\boldsymbol{g}, E).
\end{aligned}
$$

For the third $\geq$, note that the $(1 - \frac{1}{k^*})^{k^*}$ is monotonically increasing, and its limit is $\frac{1}{e}$, and $\frac{\hat{k}}{k^*} \geq 0$.

$\square$

**Lemma 2.** *Let $\boldsymbol{\omega}$ and $\boldsymbol{\omega}^* \in \mathbb{R}$ s.t. $\|\boldsymbol{\omega}^*\|_0^{\mathcal{G}} \leq k^*$. Let $S = supp(P_{\hat{k}}^{\mathcal{G}}(\boldsymbol{\omega})), S^* = supp(\boldsymbol{\omega}^*)$ and $M = S^* \setminus S$. Then the following holds*

$$
\frac{\mathbf{DIS}^*(\boldsymbol{\omega}, M)}{k^*} - \frac{\xi}{k - \hat{k}} \leq \frac{\mathbf{DIS}^*(\boldsymbol{\omega}, S \setminus S^*)}{k - \hat{k}},
$$

*where $\hat{k} = O(k^* log(\|w^*\|/\xi))$, if $\mathcal{G}$ is non-overlapped , then $\xi = 0$ .*

*Proof.* Let $O = \{G_{i_1}, G_{i_2}, \dots, G_{i_k}\}$ be the $k$-groups selected when Algorithm 2 is applied to $\boldsymbol{\omega}$, and let $Q = \{i_1, i_2, \cdots, i_k\}$. Then,

$$
\mathbf{DIS}^*(\boldsymbol{\omega}, G_{i_j} \setminus (\cup_{1 \leq \ell \leq j-1} G_{i_\ell})) \geq \mathbf{DIS}^*(\boldsymbol{\omega}, G_i \setminus (\cup_{1 \leq \ell \leq j-1} G_{i_\ell})), \quad \forall 1 \leq j \leq k, \quad \forall i \notin Q.
$$

Now, as $\cup_{1 \leq \ell \leq j-1} G_{i_\ell} \subseteq S, \forall 1 \leq j \leq k$, we have

$$
\mathbf{DIS}^*(\boldsymbol{\omega}, G_{i_j} \setminus (\cup_{1 \leq \ell \leq j-1} G_{i_\ell})) \geq \mathbf{DIS}^*(\boldsymbol{\omega}, G_i \setminus S), \quad \forall 1 \leq j \leq k, \quad \forall i \notin Q.
$$

Let $P = \{\ell_1, \dots, \ell_{k^*}\}$, $s.t.$ $supp(\boldsymbol{\omega}) \subset \cup_{1 \leq j \leq k^*} G_{\ell_j}$. Then, adding the above inequalities for all $\ell_j$ s.t. $\ell_j \in P$, we get

$$
\mathbf{DIS}^*(\boldsymbol{\omega}, G_{i_j} \setminus (\cup_{1 \leq \ell \leq j-1} G_{i_\ell})) \geq \frac{\mathbf{DIS}^*(\boldsymbol{\omega}, S^* \setminus S)}{k^*}, \tag{11}
$$

where the above inequality also uses the fact that $\sum_{\ell_j \in P} \mathbf{DIS}^*(\boldsymbol{\omega}, G_{\ell_j} \setminus S) \geq \mathbf{DIS}^*(\boldsymbol{\omega}, S^* \setminus S)$. Adding (11) $\forall (\hat{k} + 1) \leq j \leq k$, we can get

$$
\mathbf{DIS}^*(\boldsymbol{\omega}, S) - \mathbf{DIS}^*(\boldsymbol{\omega}, B) \geq \frac{k - \hat{k}}{k^*} \cdot \mathbf{DIS}^*(\boldsymbol{\omega}, S^* \setminus S),
$$

where $B = \cup_{1 \leq j \leq \hat{k}} G_{i_j}$. Moreover using Lemma 1, we get

$$
\mathbf{DIS}^*(\boldsymbol{\omega}, B) \geq (1 - e^{-\frac{\hat{k}}{k^*}}) \mathbf{DIS}^*(\boldsymbol{\omega}, S^*),
$$

and if we set $\hat{k} = O(k^* \log(\|w^*\|_2/\xi))$ (note: $\mathbf{DIS}^*(\boldsymbol{\omega}, S^*) \le \|w_{S^*}\|^2$)

$$\frac{\mathbf{DIS}^*(\boldsymbol{\omega}, M)}{k^*} \le \frac{\mathbf{DIS}^*(\boldsymbol{\omega}, S) - \mathbf{DIS}^*(\boldsymbol{\omega}, B)}{k - \hat{k}}$$

$$\le \frac{\mathbf{DIS}^*(\boldsymbol{\omega}, S) - \mathbf{DIS}^*(\boldsymbol{\omega}, S^*) + \xi}{k - \hat{k}}$$

$$\le \frac{\mathbf{DIS}^*(\boldsymbol{\omega}, S \setminus S^*) + \xi}{k - \hat{k}},$$

and if $\mathcal{G}$ is non-overlapped $\xi = 0$. Lemma now follows by a simple manipulation of the above-given inequality.

$\square$

**Lemma 3.** *Under the Assumption1 and Assumption2 we can obtain, for all $\|x\|_0^{\mathcal{G}} \le k + k^*$ the following inequality must hold*

$$\mathbb{E}_{u_i \sim N(0,\boldsymbol{I})} \|g(x) - \nabla f(x)\| = \mathbb{E}_{u_i \sim N(0,\boldsymbol{I})} \|(\nabla F(x) - \nabla f(x)) + (g(x) - \nabla F(x))\|$$

$$\le \mathbb{E}_{u_i \sim N(0,\boldsymbol{I})} \|\nabla F(x) - \nabla f(x)\| + \mathbb{E}_{u_i \sim N(0,\boldsymbol{I})} \|g(x) - \nabla F(x)\|$$

$$\le \sqrt{d} L_{k+k^*} \sigma + \frac{\sqrt{d}B}{\sqrt{n}\sigma}.$$

*Proof.* For the first part see (Berahas et al., 2022), then we proof the second part. We define

$$g(x) = \frac{1}{2n} \sum_{i=1}^{n} \left(\frac{f(x + \sigma u_i) - f(x - \sigma u_i)}{\sigma}\right) u_i$$

$$= \frac{1}{2n} \sum_{i=1}^{n} \frac{f(x + \sigma u_i)}{\sigma} u_i + \frac{1}{2n} \sum_{i=1}^{n} \frac{f(x - \sigma u_i)}{\sigma} (-u_i)$$

$$= \frac{1}{2}(g^+(x) + g^-(x)),$$

then we have

$$\mathbb{E}_{u_i \sim N(0,\boldsymbol{I})} \|g(x) - \nabla F(x)\| \le \frac{1}{2} \mathbb{E}_{u_i \sim N(0,\boldsymbol{I})} \|g^+(x) - \nabla F(x)\| + \frac{1}{2} \mathbb{E}_{u_i \sim N(0,\boldsymbol{I})} \|g^-(x) - \nabla F(x)\|$$

$$= \mathbb{E}_{u_i \sim N(0,\boldsymbol{I})} \|g^+(x) - \nabla F(x)\|$$

$$\le (\mathbb{E}_{u_i \sim N(0,\boldsymbol{I})} \|g^+(x) - \nabla F(x)\|^2)^{\frac{1}{2}}$$

$$= (\mathbb{E}_{u_i \sim N(0,\boldsymbol{I})} \|\frac{1}{n} \sum_{i=1}^{n} (\frac{f(x + \sigma u_i)}{\sigma} u_i - \nabla F(x))\|^2)^{\frac{1}{2}}$$

$$= (\frac{1}{n} \mathbb{E}_{u_1 \sim N(0,\boldsymbol{I})} (\|\frac{f(x + \sigma u_1)}{\sigma} u_1\|^2 - \|\nabla F(x)\|^2)^{\frac{1}{2}}$$

$$\le (\frac{1}{n} \mathbb{E}_{u_1 \sim N(0,\boldsymbol{I})} (\|\frac{f(x + \sigma u_1)}{\sigma} u_1\|^2)^{\frac{1}{2}}$$

$$\le (\frac{B^2}{n\sigma^2} \mathbb{E}_{u_1 \sim N(0,\boldsymbol{I})} (\|u_1\|^2)^{\frac{1}{2}}$$

$$= \frac{\sqrt{d}B}{\sqrt{n}\sigma}.$$

$\square$

**Lemma 4.** *Q is a closed convex set, $x^* \in Q$, we can obtain*

$$\|\boldsymbol{y} - \boldsymbol{x}^*\|^2 \ge \|\mathcal{P}_Q(\boldsymbol{y}) - \boldsymbol{x}^*\|^2.$$

**Lemma 5.** *Let* $\text{supp}(p_{t+1}) = S_{t+1}, \text{supp}(\delta^*) = S_*, H = S_{t+1} \cup S_*, M = S_* \backslash S_{t+1}$, *note that* $(\tilde{\delta}_{t+1})_{s_{t+1}} = (p_{t+1})_{s_{t+1}}, \hat{k} = O(k^* \log(\|\delta^*\|/\xi))$, *set* $\eta = 1/L_{k+k^*}$ *the following recursive relationship holds:*

$$\mathbb{E}\|\delta_{t+1} - \delta^*\| \leq \rho \mathbb{E}\|\delta_t - \delta^*\| + a + b + c$$

*where* $\rho = (1 + \sqrt{\frac{k^*}{k-\hat{k}}})((1 - \frac{\alpha_{k+k^*}}{L_{k+k^*}}); a = \frac{1+\sqrt{k^*/(k-\hat{k})}}{L_{k+k^*}} \cdot (\sqrt{d}L_{k+k^*}\sigma + \frac{\sqrt{d}B}{\sqrt{n}\sigma});$
$b = \frac{1+\sqrt{k^*/(k-\hat{k})}}{L_{k+k^*}} \cdot max\{\|\nabla f(\delta^*)_G\|_2 \mid G = \cup_{i=1}^{\tilde{k}}G_{j_i}, G_{j_i} \in \mathcal{G}, \tilde{k} \leq k + k^*\}; c = \sqrt{\frac{k^*\xi}{k-\hat{k}}}.$

*Proof.*

$$\|(\Pi(p_{t+1}) - \Pi(\tilde{\delta}_{t+1}))_H\|^2 = \|(\Pi(p_{t+1}) - \Pi(\tilde{\delta}_{t+1}))_{S_{t+1} \cup (S_* \backslash S_{t+1})}\|^2$$
$$= \|(\Pi(p_{t+1}) - \Pi(\tilde{\delta}_{t+1}))_{S_{t+1}} + (\Pi(p_{t+1}) - \Pi(\tilde{\delta}_{t+1}))_{(S_* \backslash S_{t+1})}\|^2$$
$$= \|(\Pi(p_{t+1}))_M - (\Pi(\tilde{\delta}_{t+1}))_M\|^2$$
$$= \|(\Pi(\tilde{\delta}_{t+1}))_M\|^2.$$

By lemma2 and the definition of $\mathbf{DIS}^*(\cdot, \cdot)$, we can obtain that

$$\|(\Pi(\tilde{\delta}_{t+1}))_M\|^2 = \mathbf{DIS}^*(\Pi(\tilde{\delta}_{t+1}), M) \leq \frac{k^*}{k-\hat{k}}\mathbf{DIS}^*(\Pi(\tilde{\delta}_{t+1}), S_{t+1} \backslash S_*) + \frac{k^*\xi}{k-\hat{k}}$$
$$= \frac{k^*}{k-\hat{k}}\|(\Pi(\tilde{\delta}_{t+1}))_{S_{t+1} \backslash S_*}\|^2 + \frac{k^*\xi}{k-\hat{k}}$$
$$= \frac{k^*}{k-\hat{k}}\|(\delta^* - \Pi(\tilde{\delta}_{t+1}))_{S_{t+1} \backslash S_*}\|^2 + \frac{k^*\xi}{k-\hat{k}}$$
$$\leq \frac{k^*}{k-\hat{k}}\|(\delta^* - \Pi(\tilde{\delta}_{t+1}))_{S_{t+1} \cup S_*}\|^2 + \frac{k^*\xi}{k-\hat{k}}$$
$$= \frac{k^*}{k-\hat{k}}\|(\delta^* - \Pi(\tilde{\delta}_{t+1}))_H\|^2 + \frac{k^*\xi}{k-\hat{k}}.$$

so $\|(\Pi(p_{t+1}) - \Pi(\tilde{\delta}_{t+1}))_H\| \leq \sqrt{\frac{k^*}{k-\hat{k}}}\|(\delta^* - \Pi(\tilde{\delta}_{t+1}))_H\| + \sqrt{\frac{k^*\xi}{k-\hat{k}}}$, then we obtain

$$\|\delta_{t+1} - \delta^*\| = \|\Pi(p_{t+1}) - \delta^*\| = \|(\Pi(p_{t+1}) - \delta^*)_H\|$$
$$= \|(\Pi(p_{t+1}) - \Pi(\tilde{\delta}_{t+1}))_H\| + \|(\Pi(\tilde{\delta}_{t+1}) - \delta^*)_H\|$$
$$\leq (1 + \sqrt{\frac{k^*}{k-\hat{k}}})\|(\Pi(\tilde{\delta}_{t+1}) - \delta^*)_H\| + \sqrt{\frac{k^*\xi}{k-\hat{k}}}.$$

By lemma 4 and $\tilde{\delta}_{t+1} = \delta_t - \eta g_t$ then we obtain

$$\|\delta_{t+1} - \delta^*\| \leq (1 + \sqrt{\frac{k^*}{k-\hat{k}}})\|(\tilde{\delta}_{t+1} - \delta^*)_H\| + \sqrt{\frac{k^*\xi}{k-\hat{k}}}$$
$$= (1 + \sqrt{\frac{k^*}{k-\hat{k}}})\|((\delta_t - \eta g_t)) - \delta^*)_H\| + \sqrt{\frac{k^*\xi}{k-\hat{k}}}.$$

By the mean value theorem, assumption 1 and lemma 3

$$
\begin{aligned}
\mathbb{E}\|(\delta_t - \eta g_t - \delta^*)_H\| &\leq \mathbb{E}\|(\delta_t - \delta^* - \eta(\nabla f(\delta_t) - \nabla f(\delta^*)))_H\| + \eta\mathbb{E}\|(\nabla f(\delta_t) - g_t)_H\| \\
&\quad + \eta\|(\nabla f(\delta^*))_H\| \\
&\leq \mathbb{E}\|\delta_t - \delta^* - \eta(\nabla f(\delta_t) - \nabla f(\delta^*))\| + \eta\mathbb{E}\|\nabla f(\delta_t) - g_t\| \\
&\quad + \eta\|(\nabla f(\delta^*))_H\| \\
&\leq \mathbb{E}\|\delta_t - \delta^* - \eta\boldsymbol{H}(\theta\delta_t + (1-\theta)\delta^*) \cdot (\delta_t - \delta^*)\| + \eta\mathbb{E}\|\nabla f(\delta_t) - g_t\| \\
&\quad + \eta\|(\nabla f(\delta^*))_H\| \\
&\leq \mathbb{E}\|(\boldsymbol{I} - \eta\boldsymbol{H}(\theta\delta_t + (1-\theta)\delta^*))\| \cdot \|(\delta_t - \delta^*)\| + \eta\mathbb{E}\|\nabla f(\delta_t) - g_t\| \\
&\quad + \eta\|(\nabla f(\delta^*))_H\| \\
&\leq (1 - \eta\alpha_{k+k^*})\mathbb{E}\|(\delta_t - \delta^*)\| + \eta(\sqrt{d}L_{k+k^*}\sigma + \frac{\sqrt{d}B}{\sqrt{n}\sigma}) \\
&\quad + \eta\|(\nabla f(\delta^*))_H\|,
\end{aligned}
$$

where $0 \leq \theta \leq 1$. A detailed explanation of the last inequality: for a matrix $\boldsymbol{A}$, its 2-norm is defined as $\|\boldsymbol{A}\|_2 = \max\{\|\boldsymbol{A}x\|_2 \mid x \in \mathbb{R}^d, \|x\|_2 = 1\}$, and it is easy to check that the 2-norm of a positive matrix is equal to its largest eigenvalue. We set $\eta \leq \frac{1}{L_{k+k^*}}$, so $0 \preceq \boldsymbol{I} - \eta\boldsymbol{H}(\theta\delta_t + (1-\theta)\delta^*) \preceq I - \alpha_{k+k^*}\eta\boldsymbol{I}$. We assume that $\boldsymbol{x}$ is the unit eigenvector corresponding to the largest eigenvalue of matrix $\boldsymbol{I} - \eta\boldsymbol{H}(\theta\delta_t + (1-\theta)\delta^*)$

$$
\begin{aligned}
1 - \alpha_{k+k^*}\eta &= \|\boldsymbol{I} - \alpha_{k+k^*}\eta\boldsymbol{I}\| \\
&\geq \|\boldsymbol{x}\| \cdot \|(\boldsymbol{I} - \alpha_{k+k^*}\eta\boldsymbol{I})\boldsymbol{x}\| \\
&\geq \boldsymbol{x}^\intercal(\boldsymbol{I} - \alpha_{k+k^*}\eta\boldsymbol{I})\boldsymbol{x} \\
&\geq \boldsymbol{x}^\intercal(\boldsymbol{I} - \eta\boldsymbol{H}(\theta\delta_t + (1-\theta)\delta^*))\boldsymbol{x} \\
&= \|\boldsymbol{I} - \eta\boldsymbol{H}(\theta\delta_t + (1-\theta)\delta^*)\|.
\end{aligned}
$$

$\square$

It is obvious that Theorem 2 comes from Lemma 5.

## D  EXPERIMENT

In this section, we will detail the details of the experimental setup (D.1), the untargeted attack performance of all baseline algorithms, and our algorithm on the CIFAR10 and MNIST datasets (D.6), and the performance on the large dataset ImageNet (D.7).

### D.1  EXPERIMENT SETTING

The experiments were conducted using Python 3.9 (Van Rossum & Drake, 2009) on a system running Ubuntu 22.04 with 4 NVIDIA GeForce 1080Ti GPUs. We used pre-trained models from both TensorFlow (Abadi et al., 2016) and PyTorch (Paszke et al., 2019) for evaluation. All attack algorithms were implemented using Numpy (Harris et al., 2020) and PyTorch. The global random seed was set to 0 in the experiment.

### D.2  DATASETS AND MODELS

To assess the comprehensive performance of our method, we conducted experiments on three datasets: MNIST, CIFAR10, and ImageNet(Krizhevsky et al., 2017). In addition, we evaluated the performance of all baseline methods on a pre-trained Inception-v3 model and a pre-trained ViT-B/16 model from Pytorch on ImageNet respectively, and we trained a CNN model on CIFAR10 and MNIST datasets respectively. To this end, we randomly selected 100 images from the ImageNet validation set, 1000 images from the CIFAR10 test set, and 1000 images from the MNIST test set to generate our adversarial examples.

The MNIST dataset consists of 60,000 training samples and 10,000 test samples, where each sample is a $28 \times 28$ handwritten number with white letters on a black background. It contains the numbers 0 to 9(10 classes). The CIFAR10 dataset consists of 50,000 training samples and 10,000 test samples, where each sample is a $32 \times 32$ color image belonging to one of ten distinct classes. We generated adversarial examples by attacking a pre-trained Inception-v3 model (Szegedy et al., 2016), which achieves a top-1 classification accuracy of 77.45% and a top-5 classification accuracy of 96% on the ImageNet. Lastly, we evaluate the performance of our method in the Vision Transformer model, ViT-B/16 (Dosovitskiy et al., 2021), which used $224 \times 224$ as input size and $16 \times 16$ as patch size.

### D.3   EVALUATION METRICS

We utilized a range of evaluation metrics to comprehensively compare our approach's attack success rate, imperceptibility, and query efficiency. These metrics are detailed below.

1. Attack Success Rate (ASR): This metric measures the effectiveness of generating adversarial examples. It is calculated by dividing the total number of successful attack samples ($N_{adv}$) by the total number of all clean and correctly classified samples that are chosen ($N_{all}$), that is, $ASR = N_{adv}/N_{all}$.

2. Imperceptibility: We used the $\ell_p$ norm to evaluate the imperceptibility of perturbations, with lower values indicating better imperceptibility. If necessary, the $\ell_0$ and $\ell_\infty$ norms were employed as constraints in all algorithms.

3. Query Efficiency: In many cases, the attacker may be constrained by time or funding and have a limited number of queries to the classifier. Lower queries effectively reduce costs and resources. We evaluated the query efficiency of our approach by measuring the number of times the classifier is queried during the attack process.

Table 8: Window size and stride setting

|  | non-overlapping | | overlapping | |
|---|---|---|---|---|
|  | window size | stride | window size | stride |
| **CIFAR10 / MNIST** | $2 \times 2$ | 2 | $3 \times 3$ | 2 |
| **ImageNet(Inception-v3)** | $13 \times 13$ | 13 | $13 \times 13$ | 10 |
| **ImageNet(ViT-B/16)** | $14 \times 14$ | 14 | $14 \times 14$ | 10 |

Table 9: Parameter setting. Constraints are shared among algorithms that use these constraints.

| Parameter | Meaning |
|---|---|
| $\varepsilon$ | Maximum disturbance radius of each pixel point |
| $k$ | Maximum group sparsity or pixel sparsity |
| $n$ | Number of Sampling in each gradient estimation |
| $\eta$ | Max learning rate |
| $\sigma$ | Search variance |

### D.4   METHODS SETTING

Experimental Settings of all baseline methods are as follows. Detailed Settings and parameter descriptions of our methods are shown in Tab. 8 and Tab. 9.

1. Global Attack Mode: For Square Attack, we set $p = 0.05$ for the attack with $\ell_\infty$ constraint. For the Parsimonious attack, we set the initial block size to 4 for the CIFAR10 and MNIST datasets and 32 for the ImageNet datasets. Besides, we fix the mini-batch size to 64. For ZO-NGD, we set $\mu = 1$, $\gamma = 0.01$ for all datasets.

2. Region-wise Attack Mode: In this mode, we take into consideration the structure of the image, which is particularly relevant in image classification tasks. To focus on the image

structure, we select the central region of each image for the attack. In Appendix E.1, we provide additional results for perturbations in other locations. Our grouping method is to divide groups from (Xu et al., 2018) by a fixed sliding window. It can generate different perturbations by adjusting the window size and step size. The details of the specific window size and stride are in Appendix E.2. In the Sparse-RS attack, we set $\alpha = 0.1$ for MNIST and CIFAR10 datasets for all attacks. Set $\alpha = 0.1$ for targeted attacks $\alpha = 0.3$ for untargeted attacks on ImageNet datasets.

3. Pixel-wise Attack Mode: For SZOHT, we set the learning rate to $0.01$, and the population of gradient estimation is 100. We can adjust $\ell_0^{\mathcal{G}}$ to maintain the same sparsity with SZOHT. In the experiments, we set the perturb rate as $10\%$ while setting $\ell_0$ to 78 for MNIST, 307 for CIFAR10, 26820 for Inception-v3, and 15052 for ViT-B/16, respectively.

## D.5 NETWORK ARCHITECTURE

We present the network architectures for CIFAR-10 and MNIST datasets. Tab. 10 displays the network architecture for CIFAR-10, while Tab. 11 provides the network architecture for MNIST. For the model of the ImageNet dataset, we adopted the inception-v3 model provided by Pytorch's official website.

Table 10: Illustration of CIFAR-10 network architecture.

| type | kernel size / output feature | input size |
|---|---|---|
| Conv | $3{\times}3$ | $32{\times}32{\times}3$ |
| Relu | – | $32{\times}32{\times}3$ |
| Conv | $3{\times}3$ | $30{\times}30{\times}64$ |
| Relu | – | $30{\times}30{\times}64$ |
| MaxPooling | $2{\times}2$ | $28{\times}28{\times}64$ |
| Conv | $3{\times}3$ | $14{\times}14{\times}128$ |
| Relu | – | $14{\times}14{\times}128$ |
| Conv | $3{\times}3$ | $12{\times}12{\times}128$ |
| Relu | – | $12{\times}12{\times}128$ |
| MaxPooling | $2{\times}2$ | $10{\times}10{\times}128$ |
| Dense | 256 | 3200 |
| Relu | – | 256 |
| Dense | 256 | 256 |
| Relu | – | 256 |
| Dense | 10 | 256 |
| Softmax | – | 10 |

Table 11: Illustration of MNIST network architecture.

| type | kernel size / output feature | input size |
|---|---|---|
| Conv | $3{\times}3$ | $28{\times}28{\times}1$ |
| Relu | – | $28{\times}28{\times}1$ |
| Conv | $3{\times}3$ | $26{\times}26{\times}32$ |
| Relu | – | $26{\times}26{\times}32$ |
| MaxPooling | $2{\times}2$ | $24{\times}24{\times}32$ |
| Conv | $3{\times}3$ | $12{\times}12{\times}64$ |
| Relu | – | $12{\times}12{\times}64$ |
| Conv | $3{\times}3$ | $10{\times}10{\times}64$ |
| Relu | – | $10{\times}10{\times}64$ |
| MaxPooling | $2{\times}2$ | $8{\times}8{\times}64$ |
| Dense | 200 | 1024 |
| Relu | – | 200 |
| Dense | 200 | 200 |
| Relu | – | 200 |
| Dense | 10 | 200 |
| Softmax | – | 10 |

## D.6 UNTARGETED ATTACK ON CIFAR10 AND MNIST

In this section, we evaluate the untargeted attack performance of all experiments on CIFAR10 and MNIST datasets. Tab. 12 shows the performance between our method and global attack methods. We can see that our average and median queries are similar to those of other algorithms under the same amount of perturbations. At 30% of the perturbation volume, we were able to maintain a high ASR and lower detectability, despite sacrificing some queries, but these sacrifices were bearable. In Tab. 13, our performance is on par with Patch-RS. In Tab. 14, the accuracy of our algorithm far exceeds theirs under the same constraints. From Fig. 7, we can clearly see the adversarial samples and perturbations generated by all the algorithms. Compared to global methods, the perturbations generated by our method are more sparse and structured. Patch-RS drew an obvious patch on the image, and although the disturbance range can be controlled by constraints, it is very easy to detect. And, obviously, the smaller the disturbance range, the lower the success rate. For region-fixed methods, focuses on a small area, which is less flexible than Patch-RS.

Table 12: Comprehensive comparison of global attack algorithms with $\ell_\infty$ constraints on CIFAR10 and MNIST, where $\varepsilon = 0.4$ in MNIST, $\varepsilon = 0.1$ in CIFAR10.

| | | CIFAR10 | | | | | MNIST | | | |
|---|---|---|---|---|---|---|---|---|---|---|
| **Algorithm** | **ASR** | **Avg.** | **Med.** | $\ell_0$ | $\ell_2$ | **ASR** | **Avg.** | **Med.** | $\ell_0$ | $\ell_2$ |
| Parsimonious | **100.00%** | 91.2 | 82.0 | 3063.5 | 5.4 | **100.00%** | 36.4 | 38.0 | 260.8 | 6.5 |
| Square$\ell_\infty$ | **100.00%** | **13.0** | **3.0** | 3054.3 | 5.4 | **100.00%** | **34.3** | **13.0** | 459.8 | 8.1 |
| ZO-NGD | 82.60% | 2489.3 | 707.0 | 3055.8 | 5.4 | 99.30% | 469.6 | 101.0 | 469.9 | 8.2 |
| Ours(N)$_{100\%d}$ | **100.00%** | 52.4 | 9.0 | 3054.8 | 5.2 | **100.00%** | 42.6 | 90.0 | 506.7 | 7.8 |
| Ours(O)$_{100\%d}$ | **100.00%** | 63.5 | 21.0 | 3062.0 | 4.6 | **100.00%** | 39.7 | 83.5 | 557.5 | 8.0 |
| Ours(N)$_{30\%d}$ | **100.00%** | 290.1 | 64.0 | **923.4** | **3.0** | 99.47% | 392.0 | 161.0 | **191.3** | **5.3** |
| Ours(O)$_{30\%d}$ | **100.00%** | 344.2 | 65.5 | **911.7** | **2.9** | 98.06% | 736.4 | 215.5 | **175.4** | **4.9** |

[*] (N) Non-overlapping groups; (O) Overlapping groups; Number%d: the proportion of perturbed image features.

Table 13: Comprehensive comparison of Sparse-RS (Patch-RS) algorithms with $\ell_0$ constraints on MNIST and CIFAR10. The perturbation ratio of the image is 10% of all features for all algorithms.

| | | CIFAR10 | | | | | MNIST | | | |
|---|---|---|---|---|---|---|---|---|---|---|
| **Algorithm** | **ASR** | **Avg.** | **Med.** | $\ell_2$ | $\ell_\infty$ | **ASR** | **Avg.** | **Med.** | $\ell_2$ | $\ell_\infty$ |
| Patch-RS | **100.00%** | **16.7** | 7.0 | 9.6 | 0.9 | **100.00%** | **30.8** | 12.0 | **7.0** | 1.0 |
| Ours(N)$_{\epsilon=1}$ | **100.00%** | 39.5 | **6.0** | 10.1 | 0.9 | **100.00%** | 90.7 | 60.0 | 8.5 | 1.0 |
| Ours(O)$_{\epsilon=1}$ | **100.00%** | 33.5 | **3.0** | **8.8** | 0.9 | **100.00%** | 36.9 | **15.0** | 9.4 | 1.0 |

[*] (N) Non-overlapping groups; (O) Overlapping groups.

Table 14: Comprehensive comparison of region-fixed targeted attack algorithms with $\ell_{0+\infty}$ constraints on MNIST and CIFAR10, where $\varepsilon = 0.4$ in MNIST, $\varepsilon = 0.1$ in CIFAR10. The perturbation ratio of the image is 10% of all features for all algorithms.

| | | CIFAR10 | | | MNIST | |
|---|---|---|---|---|---|---|
| **Algorithm** | **ASR** | **Avg.** | **Med.** | **ASR** | **Avg.** | **Med.** |
| Fixed-Parsimonious | 74.00% | 2871.5 | 349.0 | 40.44% | 6009.6 | 10000.0 |
| Fixed-Square$\ell_\infty$ | 72.04% | 3079.4 | **75.0** | 54.72% | 4601.6 | **259.0** |
| Fixed-ZO-NGD | 48.60% | 5748.7 | 10000.0 | 45.80% | 7965.9 | 10000.0 |
| Ours(N)$_{10\%d}$ | **96.10%** | **1291.9** | 450.5 | **94.56%** | **1310.5** | 385.0 |
| Ours(O)$_{10\%d}$ | **95.71%** | **1428.3** | 480.5 | **87.45%** | **1856.4** | 415.5 |

[*] (N) Non-overlapping groups; (O) Overlapping groups; Number%d: the proportion of perturbed image features.

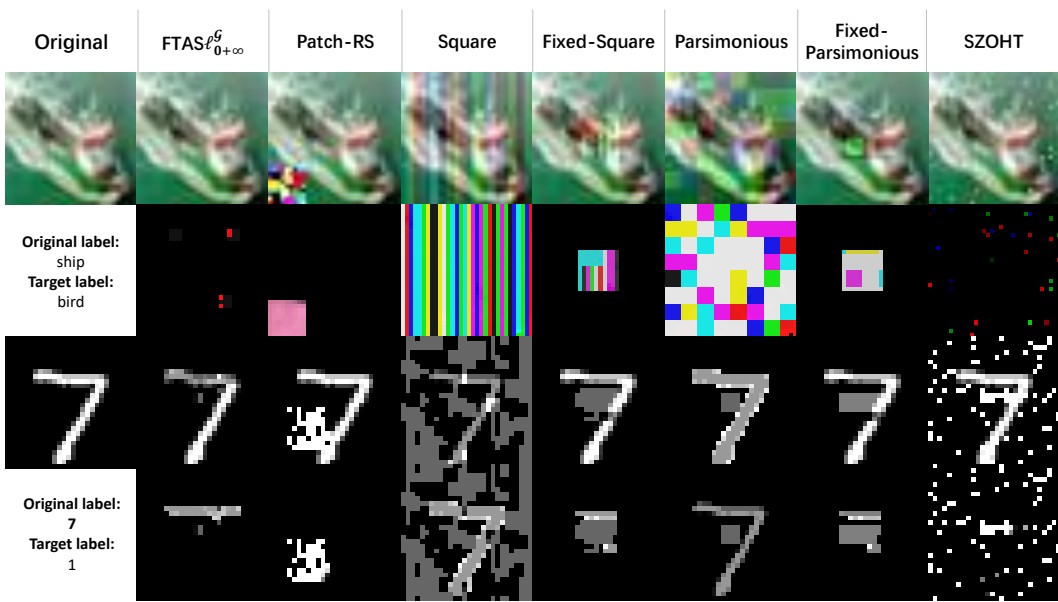

Figure 7: Visualization results on the CIFAR10 and MNIST dataset.

## D.7 PERFORMANCE ON IMAGENET

In this section, we investigate the effectiveness of untargeted attacks on the ImageNet dataset. We comprehensively evaluate three modes including global attack, region-wise attack, and region-fixed attack. The region-fixed attack method is the area of the attack fixed in the center of the image. Through our extensive experiments, we uncover a significant decline in the performance of region-fixed algorithms when applied to the ImageNet dataset. Notably, our method demonstrates a higher success rate and requires fewer queries compared to other approaches, while maintaining the same level of perturbation and query limit. Furthermore, Fig. 8 provides visual evidence of our findings. By analyzing the figure, we observe more pronounced structural perturbations in the targeted regions of the images, indicating the effectiveness and precision of our method in generating adversarial examples on the ImageNet dataset.

Table 15: Comprehensive comparison of global attack algorithms with $\ell_\infty$ constraints on Inception-v3 and ViT-B/16 model, ImageNet dataset, where $\varepsilon = 0.1$.

| Algorithm | Inception-v3 | | | | VIT | | | |
| | ASR | Avg. | Med. | $\ell_0$ | ASR | Avg. | Med. | $\ell_0$ |
|---|---|---|---|---|---|---|---|---|
| Parsimonious | 98.93% | 1062.2 | 500.5 | 265273.3 | **100.00%** | 761.1 | 489.5 | 149019.5 |
| Square$\ell_\infty$ | **100.00%** | **49.3** | **21.0** | 264788.7 | **100.00%** | **29.3** | **19.0** | 148679.6 |
| ZO-NGD | 72.97% | 6180.0 | 600.0 | 265116.9 | 84.00% | 4469.0 | 500.0 | 142752.4 |
| Our(N)$_{100\%d}$ | **100.00%** | 329.1 | 143.0 | 264985.3 | **100.00%** | 447.0 | 182.0 | 148934.4 |
| Our(O)$_{100\%d}$ | **100.00%** | 98.4 | 51.5 | 265397.0 | **100.00%** | 483.0 | 212.5 | 149007.7 |
| Our(N)$_{30\%d}$ | 92.63% | 3910.8 | 1302.0 | **80307.4** | 98.48% | 3166.8 | 1502.0 | **45099.1** |
| Our(O)$_{30\%d}$ | 96.88% | 2811.1 | 587.5 | **79900.2** | **100.00%** | 2523.7 | 1766.0 | **44968.8** |

[*] (N) Non-overlapping groups; (O) Overlapping groups; Number%d: the proportion of perturbed image features.

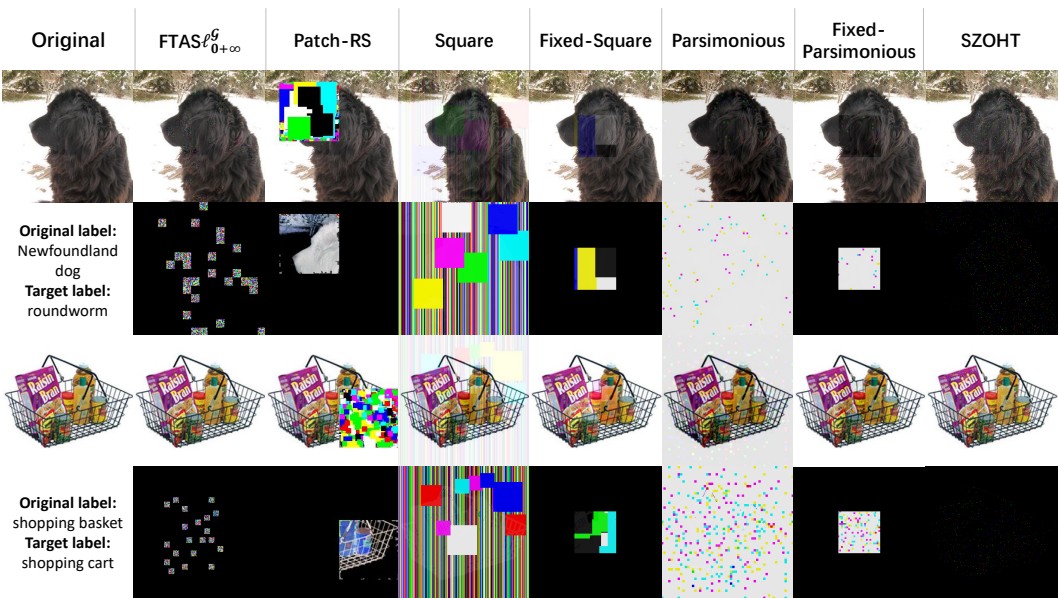

Figure 8: Visualization results on the ImageNet dataset.

Table 16: Comprehensive comparison of Sparse-RS (Patch-RS) algorithms with $\ell_0$ constraints on ImageNet. The perturbation ratio of the image is 10% of all features for all algorithms.

| Algorithm | Inception-v3 | | | | VIT | | | |
|---|---|---|---|---|---|---|---|---|
| | ASR | Avg. | Med. | $\ell_\infty$ | ASR | Avg. | Med. | $\ell_\infty$ |
| Patch-RS | 92.29% | 2968.6 | 685.5 | 1.0 | 86.90% | **3572.8** | 1849.5 | 1.0 |
| Our(N)$_{\epsilon=1}$ | **96.84%** | 3033.9 | **325.0** | 1.0 | **98.79%** | 3744.9 | **1542.0** | 1.0 |
| Our(O)$_{\epsilon=1}$ | **97.89%** | **1935.2** | 427.0 | 1.0 | **99.84%** | 4031.8 | **1106.0** | 1.0 |
| Our(N)$_{\epsilon=0.2}$ | 90.53% | 7069.2 | 2118.0 | **0.2** | 96.84% | 5120.5 | 2060.5 | **0.2** |
| Our(O)$_{\epsilon=0.2}$ | 91.53% | 7258.9 | 1571.0 | **0.2** | 92.63% | 5602.1 | 3507.5 | **0.2** |

[*] (N) Non-overlapping groups; (O) Overlapping groups.

Table 17: Comprehensive comparison of region-fixed targeted attack algorithms with $\ell_{0+\infty}$ constraints on ImageNet, where $\varepsilon = 0.1$ in ImageNet. The perturbation ratio of the image is 10% of all features for all algorithms.

| Algorithm | Inception-v3 | | | VIT | | |
|---|---|---|---|---|---|---|
| | ASR | Avg. | Med. | ASR | Avg. | Med. |
| Fixed-Parsimonious | 74.46% | 9631.3 | 7543.0 | 85.36% | 8248.5 | 4952.5 |
| Fixed-Square$\ell_\infty$ | 75.53% | 8893.3 | **248.0** | 78.95% | 8624.8 | **175.0** |
| Fixed-ZO-NGD | 79.12% | 7436.0 | 1500.0 | 80.98% | 9426.0 | 3700.0 |
| Our(N)$_{10\%d}$ | 74.21% | 9003.4 | 7261.0 | 74.74% | 8495.4 | 6838.0 |
| Our(O)$_{10\%d}$ | **81.88%** | **7157.8** | 4878.0 | **88.21%** | **8182.8** | 5971.5 |

[*] (N) Non-overlapping groups; (O) Overlapping groups; Number%d: the proportion of perturbed image features.

# E ABLATION EXPERIMENT

## E.1 REGION-FIXED IN DIFFERENT LOCATION

To comprehensively demonstrate the exceptional efficiency of our algorithm, we present additional results involving perturbations in various locations, namely upper left, upper right, lower left, and lower right. These results serve as a supplement to the central location analysis discussed in the main text. Tab. 18 and Tab. 19 showcases the Attack Success Rate (ASR) and Avgerage queries of Fixed-Parsimonious, Fixed-Square, and Fixed-ZO-NGD at these different locations. We randomly selected 1000 examples from the CIFAR10 and MNIST datasets and 100 examples from the ImageNet dataset for untargeted attacks. Consistent with our expectations, the attack achieves better performance when targeted at the center of the image compared to almost any other location.

Table 18: Comparsion at different fixed locations on CIFAR10 and MNIST datasets

| Algorithm | Location | CIFAR10 | | MNIST | |
|---|---|---|---|---|---|
| | | ASR | Avg. queries | ASR | Avg. queries |
| **Fixed-Parsimonious** | upper left | 26.79% | 7486.00 | 1.04% | 9944.52 |
| | upper right | 34.57% | 6717.46 | 2.09% | 9842.59 |
| | lower left | 32.93% | 6937.69 | 2.60% | 9912.50 |
| | lower right | 29.08% | 7259.23 | 1.56% | 9902.14 |
| | center | **74.00%** | **2871.53** | **40.44%** | **6009.62** |
| **Fixed-Square** | upper left | 49.11% | 5990.96 | 0.85% | 9914.76 |
| | upper right | 47.84% | 6143.28 | 2.77% | 9733.05 |
| | lower left | 52.42% | 5776.13 | 0.85% | 9915.71 |
| | lower right | 51.15% | 5705.99 | 1.49% | 9852.38 |
| | center | **72.04%** | **3079.37** | **54.72%** | **4601.62** |
| **Fixed-ZO-NGD** | upper left | 17.00% | 8517.15 | 39.50% | 7952.84 |
| | upper right | 26.80% | 7701.17 | 31.20% | 8546.25 |
| | lower left | **50.00%** | **5665.59** | 30.97% | 8715.36 |
| | lower right | 37.60% | 6849.63 | 27.69% | 9812.94 |
| | center | 48.60% | 5748.72 | **45.80%** | **7965.90** |

Table 19: Comparsion at different fixed locations on ImageNet datasets

| Algorithm | Location | Inceptionv3 | | ViT-B/16 | |
|---|---|---|---|---|---|
| | | ASR | Avg. queries | ASR | Avg. queries |
| **Fixed-Parsimonious** | upper left | 22.66% | 17185.00 | 69.73% | 10861.19 |
| | upper right | 20.00% | 17212.49 | 71.94% | 9483.68 |
| | lower left | 17.20% | 16730.18 | 17.10% | 16673.15 |
| | lower right | 18.66% | 16504.45 | 15.78% | 16903.11 |
| | center | **74.46%** | **9631.30** | **74.46%** | **9631.30** |
| **Fixed-Square** | upper left | 49.33% | 13225.44 | **82.11%** | 5510.82 |
| | upper right | 50.67% | 11919.18 | 72.11% | 5339.93 |
| | lower left | 52.00% | 12452.98 | 77.37% | 4115.26 |
| | lower right | 42.67% | 13238.80 | 77.37% | **3279.78** |
| | center | **75.53%** | **5893.32** | 75.53% | 8893.30 |
| **Fixed-ZO-NGD** | upper left | 24.29% | 15685.71 | 54.29% | 10431.42 |
| | upper right | 38.75% | 14637.50 | 68.75% | 9880.00 |
| | lower left | 43.75% | 13346.25 | 67.50% | 10062.50 |
| | lower right | 38.75% | 14506.25 | 71.25% | **8977.50** |
| | center | **79.12%** | **7436.00** | **79.12%** | 9003.40 |

## E.2 GROUPING

We conducted a comprehensive untargeted attack experiment, varying the window sizes and strides. Our experiment involved 500 samples from the CIFAR10 and MNIST datasets, as well as 80 samples from the ImageNet dataset. Additionally, we ensured that each sample retained a 10% level of perturbation and attacked the Inception-v3 model for the ImageNet dataset.

Tab. 20 presents the results of our experiment, highlighting the impact of different filter sizes on the attack performance. From the table, it is evident that the choice of filter size significantly influences the effectiveness of the attack in MNIST and ImageNet datasets. A smaller window size allows for more precise refinement of the target structure, resulting in better-formed and structured perturbations. In contrast, larger windows tend to lose sensitivity to the intricate details of the image structure. Consequently, the choice of window size directly affects the level of refinement and the overall quality of the perturbations generated during the attack.

Table 20: Comparsion of different grouping

| | | | Non-overlapping | | | | Overlapping | | |
| --- | --- | --- | --- | --- | --- | --- | --- | --- | --- |
| | FilterSize | Stride | k | ASR | FilterSize | Stride | k | ASR |
| CIFAR10 | 2 | 2 | 20 | **99.63%** | 3 | 2 | 11 | **99.63%** |
| | 4 | 4 | 5 | 98.63% | 5 | 3 | 4 | 98.63% |
| | 6 | 6 | 3 | 95.89% | 7 | 5 | 2 | 93.15% |
| MNIST | 2 | 2 | 20 | **92.69%** | 3 | 2 | 10 | **93.01%** |
| | 4 | 4 | 5 | 86.39% | 5 | 3 | 3 | 87.24% |
| | 6 | 6 | 3 | 65.77% | 7 | 5 | 2 | 65.77% |
| ImageNet | 13 | 13 | 80 | 82.69% | 13 | 10 | 80 | **94.23%** |
| | 15 | 15 | 60 | 86.27% | 15 | 10 | 60 | 92.31% |
| | 20 | 20 | 33 | **88.46%** | 20 | 15 | 33 | 86.54% |

# F  SUPPLEMENTARY

## F.1  VALIDATION ABOUT APPROXIMATE SOLUTION

The theory and time above suggest that as the number of selected groups increases, the discrepancy between approximate and optimal solutions diminishes accordingly. To validate this lemma, we conduct an experimental verification in this section. We define the optimal solution as a k-group sparse selection. The image dimensions are (1, 40, 40), the filter size is 10, and the stride value is set to 4. Consequently, we obtained 64 overlapping groups, with each group consisting of 100 points.

Fig. 9 provides a graphical representation of the results from the validation experiment. We vary the value of $k$ to be 10, 20, 30, and 40, respectively. As $\hat{k}$ increases, we observe a gradual convergence of the gap between the approximate and optimal solutions towards an upper bound. By observing the figure, it is evident that during the initial stages of the projection process, the gap between the approximate and optimal solutions decreases rapidly. As more groups are selected, the proportion of non-overlapping parts in the remaining groups becomes smaller, resulting in a slower rate of gap reduction. This pattern indicates that the impact of each subsequent group selection becomes less significant as the algorithm progresses, leading to a gradual convergence of the gap toward a m:

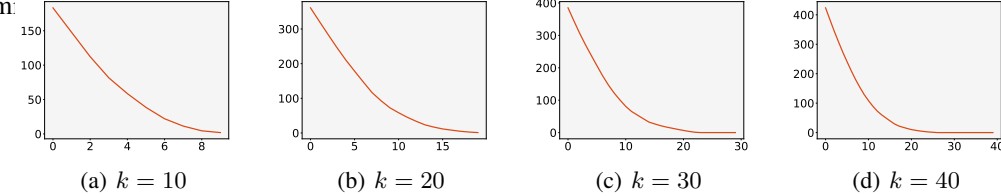

(a) $k = 10$      (b) $k = 20$      (c) $k = 30$      (d) $k = 40$

Figure 9: Validation experiment w.r.t different k.

## F.2  QUERY DISTRIBUTION

We have included additional query distribution histograms for the CIFAR10, MNIST, and ImageNet datasets in Fig. 10. This addition expands the scope of our analysis and provides insights into the distribution patterns of queries generated by our algorithm. Notably, we can observe that our algorithm can generate a majority of adversarial examples using a relatively low number of queries. This finding is particularly relevant in scenarios where there are limitations on the query budget. It suggests

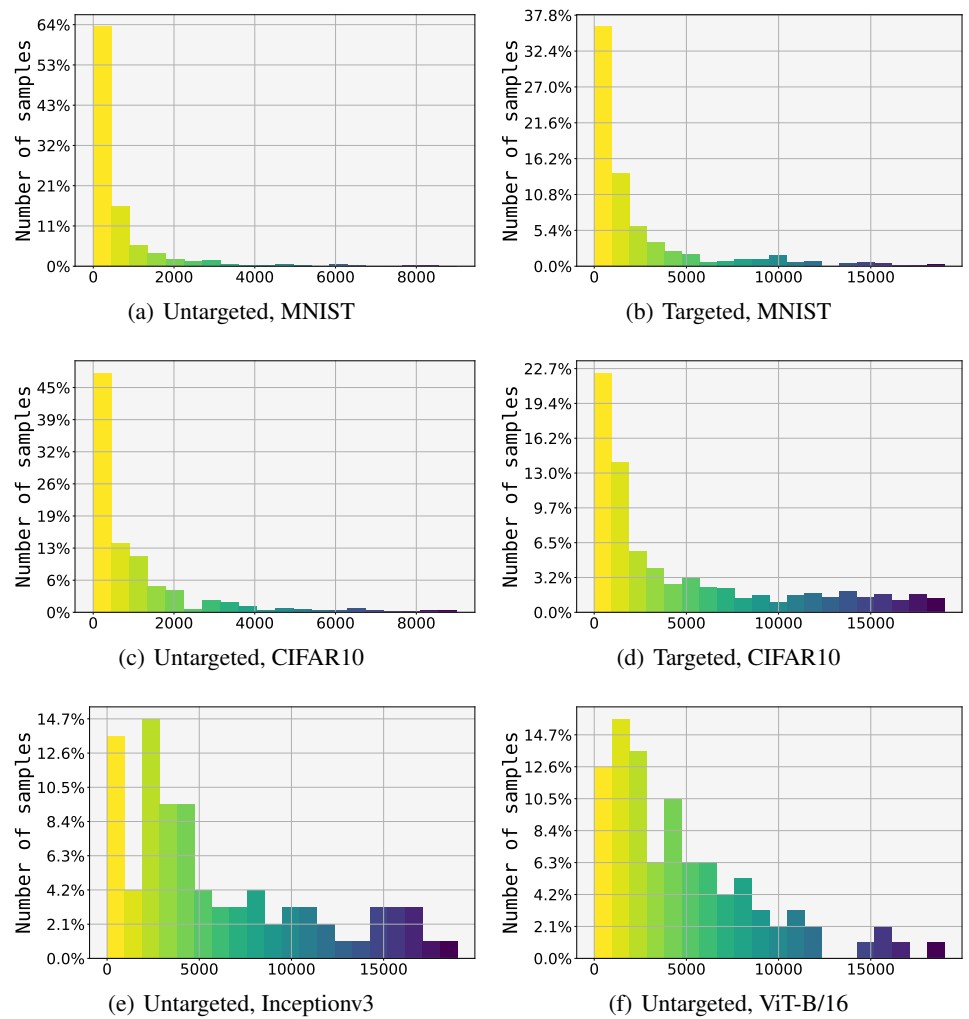

Figure 10: Query distribution. From top to bottom, the first row is the MNIST, where $\varepsilon = 0.4$ and the second row is the CIFAR10, where $\varepsilon = 0.1$ and the last row is the ImageNet, where $\varepsilon = 0.1$. For all graphs, $k \approx 10\%d$.

that our algorithm can be implemented effectively within various query budget constraints while also reducing the likelihood of being detected by defenses designed to detect numerous queries.

### F.3    DIFFERENT QUERY BUDGET

We set different query budget that is 5,000, 10,000, 15,000, 20,000, and 25,000 instead of a fixed budget. Notwithstanding the inherent disparities in our respective problems, we divided three sets of experiments for different problems (Global attack, Region-wise attack, Fixed attack). First, we compared with global attack methods, i.e. Square attack, Parsimonious attack, and ZO-NGD attack, with $\ell_\infty$ constraints in Fig. 11(a,b). Thus, we set the same $\ell_\infty$ constraint for all algorithms is 0.05. Second, we compared the region-wise attack, i.e. Patch-RS, with a patch in Fig. 11(c,d). We set the patch size of the patch to 80 in Patch-RS. The $\ell_0$ constraint is 19200($80 \times 80 \times 3$). Finally, we compared with the fixed-version of global attack algorithms in Fig. 11(e,f). For all algorithms, we set the same double constraint, $\ell_\infty$ is 0.1, $\ell_0$ is 26820 for the ImageNet dataset, Inceptionv3 model, and $\ell_\infty$ is 0.1, $\ell_0$ is 15052 for ImageNet dataset, vision transformer model. That is, for double constraint, we perturb only 10% of the pixels.

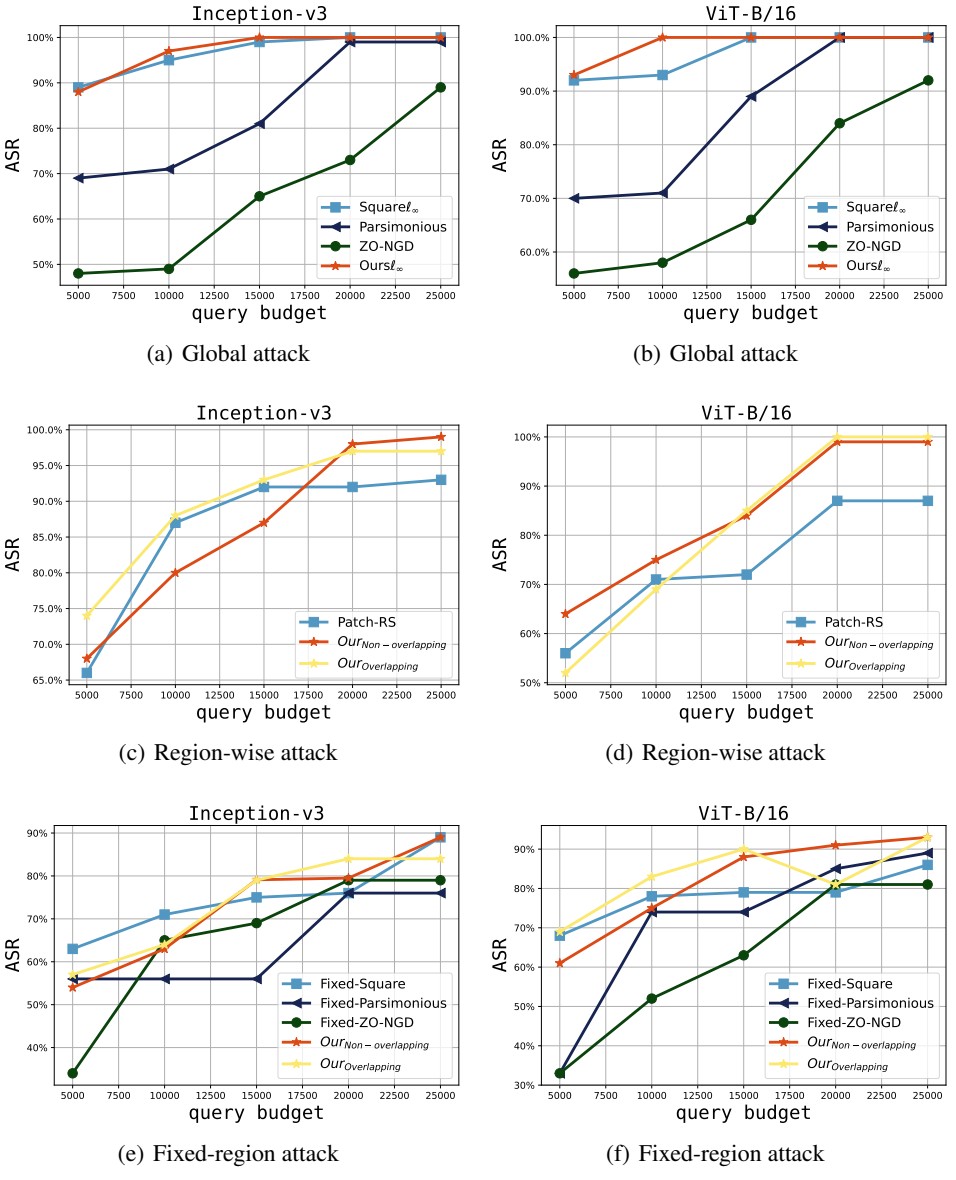

Figure 11: Demonstrate the attack strength of all algorithms under different query budgets.

From the Fig. 11(a,b), all algorithms have $\ell_\infty$ constraint 0.05. We can see that the success rate of the attack will decrease significantly if the query is less than 15,000 for high-resolution image attack tasks. When the query budget reaches 20,000, the success rate becomes stable. From Fig. 11(c,d), all algorithms perturb up to 19200 of the pixels. Patch-RS is a heuristic attack method, it can be seen from Fig. 11(d) ViT-B model that Patch-RS performs very poorly under a strict query budget. From Fig 11(e,f), global attack algorithms with fixed regions perform poorly under different budgets.

## F.4 SUPPLEMENTAL EXPERIMENTS ON CIFAR10

We evaluated the CIFAR10 dataset on two models, resnet18 and mobilenet-v2. The results of the evaluation are shown in the table below. In this evaluation, we also adopted variance reduction technology, and its query efficiency and attack success rate have been effectively improved.

Table 21: Performance of CIFAR10 dataset on Resnet18 and Mobilenet-v2 model.

| Model | Attack Type | Overlap Condition | ASR | Avg. | Med. | $\ell_2$ |
|---|---|---|---|---|---|---|
| Resnet18 | untargeted | non-overlapping | 99.59% | 1249.0 | 315.0 | 1.64 |
| | | overlapping | 95.34% | 1778.3 | 129.0 | 1.72 |
| | targeted | non-overlapping | 74.93% | 8081.9 | 5115.0 | 1.81 |
| | | overlapping | 88.41% | 6040.4 | 4171.5 | 1.72 |
| Mobilenet-v2 | untargeted | non-overlapping | 99.92% | 1455.6 | 666.5 | 1.69 |
| | | overlapping | 96.73% | 4679.0 | 927.0 | 1.70 |
| | targeted | non-overlapping | 88.06% | 6318.7 | 1246.5 | 1.84 |
| | | overlapping | 85.32% | 8278.0 | 6502.5 | 1.73 |

## F.5 MULTIPLE SUBREGION IN FIXED VERSION

We simply added a baseline, the modified fixed version. Detailly, we randomly generated 5 squares with the same size on the image and fixed the 5 positions to attack. In order to ensure the consistency of the experiment, we ensured that these 5 groups disturbed 10% of the dimension of the whole image. The result is shown below. Fig. 12 shows the adversarial sample generated by Fixed-Parsimonious, Fixed-Square, and Fixed-ZO-NGD. We can clearly see the disturbance generated by Fixed-Parsimonious and Fixed-Square, while Fixed-ZO-NGD based on gradient is more imperceptible. From the results in Tab. 22, it can be seen that the Fixed Square attack gets better ASR, while other algorithms do not improve significantly. It can be seen that the attack success rate is very sensitive to the location of the region, which also prompts us to automatically select the more sensitive region by optimization.

Table 22: Results with multiple fixed subregions on ImageNet dataset.

| | | ASR | Avg. | Med. | $\ell_2$ |
|---|---|---|---|---|---|
| **Fixed-Square** | Inceptionv3 | 78.79% | 5390.17 | 3608.00 | 15.58 |
| | ViT-B/16 | 87.62% | 6991.21 | 1699.50 | 11.49 |
| **Fixed-ZO-NGD** | Inceptionv3 | 69.00% | 9167.00 | 4700.00 | 7.88 |
| | ViT-B/16 | 73.00% | 9042.00 | 4900.00 | 6.12 |
| **Fixed-Parsimonious** | Inceptionv3 | 64.57% | 13321.41 | 14942.00 | 15.75 |
| | ViT-B/16 | 71.91% | 13723.41 | 13541.50 | 11.57 |

## F.6 REGULAR TERM LOSS BASELINE

To evaluate the efficacy of an image adversarial attack algorithm under $\ell_0$ and $\ell_\infty$ constraints, with the addition of an $\ell_2$ loss term. We introduce an $\ell_2$ norm-based loss term into the optimization problem. This addition aims to refine the adversarial perturbations by considering the Euclidean distance in the perturbation space. We utilize a standard image dataset CIFAR10 for evaluating the

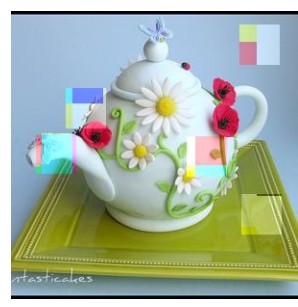 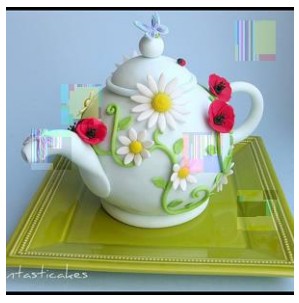 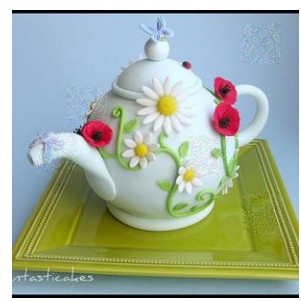

Queries: 3823          Queries: 3260          Queries: 5288

Figure 12: Visualization in the fixed version of global SOTA algorithms. From left to right are Fixed-Parsimonious, Fixed-Square, and Fixed-ZO-NGD, and these five sub-regions can be clearly seen.

adversarial attacks. We set $\ell_\infty$ constraint as 0.1, and perturb 10% pixels. We set the parameter of the penalty term, i.e. $c$ is 0.01, 0.1, 0.5, 1, 5, 10. As shown in Table 23. As $c$ increases, the $\ell_2$ decreases.

Table 23: Comparison with different $c$

| $c$ | Group type | ASR | Avg. | Med. | $\ell_2$ |
|---|---|---|---|---|---|
| 0.01 | | 97.92% | 926.06 | 439.50 | 1.67 |
| 0.1 | | 93.49% | 1449.11 | 434.50 | 1.62 |
| 0.5 | | 72.40% | 3298.92 | 699.00 | 1.31 |
| 1 | Non-overlapping | 56.25% | 4839.60 | 2481.00 | 1.04 |
| 5 | | 36.20% | 6491.54 | 10000.00 | 0.67 |
| 10 | | 33.85% | 6724.81 | 10000.00 | 0.57 |
| 0.01 | | 97.66% | 785.43 | 264.50 | 2.05 |
| 0.1 | | 92.71% | 1248.11 | 237.50 | 1.97 |
| 0.5 | | 72.40% | 3207.38 | 323.50 | 1.54 |
| 1 | Overlapping | 55.47% | 4731.00 | 1606.00 | 1.25 |
| 5 | | 40.62% | 6001.16 | 10000.00 | 0.87 |
| 10 | | 40.36% | 6018.25 | 10000.00 | 0.83 |

