# OpenReview forum: "Query Efficient  Black-Box  Adversarial Attack with Automatic Region Selection"
_ICLR.cc/2024/Conference — Submitted to ICLR 2024_

### Official Review · Reviewer_9mc2 · 2023-10-26

**Soundness:** 2 fair
**Presentation:** 2 fair
**Contribution:** 3 good
**Rating:** 5
**Confidence:** 3

**Summary:**

This paper presents FTAS, a novel algorithm for region-wise adversarial attacks in a black box setting. Existing region-wise attacks heuristically determine the perturbation region, which leads to a bad attack performance. FTAS automatically determines the perturbation region with controllable imperceptibility by solving an optimization problem that considers the perturbation region selection.
This paper provides the theoretical convergence analysis of the solution obtained by FTAS under standard assumptions.
Experimental results on different datasets indicate that FTAS requires fewer perturbations than the global-region attack and fewer queries than existing region-wise attacks. In addition, FTAS provides better interpretability on vulnerable regions, which is impossible with pixel-wise sparse attacks.

**Strengths:**

The strengths of this paper include followings.
1. This study introduces a novel formulation that incorporates region selection as a constraint in region-wise attacks and presents the FTAS algorithm.
2. FTAS outperforms existing methods, requiring fewer perturbations than global-region attacks and fewer queries than region-wise attacks.
3. FTAS offers enhanced interpretability of vulnerable region of inputs, which is impossible with pixel-wise sparse attacks.

**Weaknesses:**

Although this paper presents a new formulation for sparse and effective adversarial attacks, concerns exist regarding the gap between the problem to be solved and that actually solved, as well as the theoretical performance guarantees of the algorithm.
1. To minimize $F(x_0 + \delta, y)$ under some constraints, the authors solve the problem (5) derived from inequalities originating from the smoothness of $F$ and the Lipschitz continuity of the gradient. However, it is important to note that problem (5) does not necessarily entail the minimization of $F(x_0 + \delta, y)$.
2. This paper relies on the assumptions of RSC and RSS to provide a theoretical guarantee for the performance of Algorithm 1. Nevertheless, it should be emphasized that this assumption may not necessarily hold, particularly in the context of adversarial attacks involving complex neural networks.
3. The performance of Algorithm 1 is theoretically guaranteed by Theorem 2. However, the authors do not discuss the tightness of the bound. Notably, the right-hand side of the inequality established in Theorem 2 incorporates the variable $\rho^T$. Consequently, if $\rho>1$, the right-hand side of the inequality may diverge towards positive infinity as $T$ approaches infinity.

Additionally, minor comments:
4. Clarity issues arise in certain parts of the text due to the identical formatting of vectors and scalars.
5. The horizontal axis label in Figure 4 might require correction.
6. In the opening sentence of Section 2.2, it would be more appropriate to replace the phrase "we propose a new objective function for adversarial training" with "we propose a new problem formulation for adversarial attacks."

**Questions:**

I kindly ask the authors to answer the following questions.
1. Is the assumption of gradient Lipschitz continuity reasonable for adversarial attacks?
2. Is the assumption 1 reasonable for adversarial attacks?
3. Does theorem 2 provide meaningful performance bound of the algorithm 1 in practice?

---

> ### Author Response · Authors · 2023-11-21
>
> Q1: To minimize $F(x_+\delta, y)$ under some constraints, the authors solve the problem (5) derived from inequalities originating from the smoothness of $F$ and the Lipschitz continuity of the gradient. However, it is important to note that problem (5) does not necessarily entail the minimization of $F(x_0+\delta, y)$. Is the assumption of gradient Lipschitz continuity reasonable for adversarial attacks?
>
> A1: Thank you for your comments. $F(x_0+\delta,y)$ is gradually optimized through iteration. Instead of optimizing $F(x_0+\delta)$ directly, we optimized the subproblem which is the first-order approximation of $F(x_0+\delta,y)$ plus the regular term in each iteration. Problem (5) is a simplified version of the subproblem with double constraint. The problem (5) serves as an intermediary towards achieving the ultimate goal of minimizing $F$. Actually, problem (5) is a proximal operator, it can be written as $prox(\delta) = min \frac{L}{2}\|\delta-S_L(\delta^t)\|_2^2 + h(\delta)$, where $h(\delta)=0$ if $\|\delta\|^G_0 \leq k, l\leq \delta \leq u$, otherwise $h(\delta)=\infty$. Thus, the problem (5) is closely linked to the optimization process in adversarial attacks. From this point of view, it is necessary to minimize the problem (5). We have corrected the "minimize $F(x_0+y)$" in the paper.
>
>  Q2: Is the assumption 1 reasonable for adversarial attacks? Particularly in the context of adversarial attacks involving complex neural networks.
>
> A2: Thank you for your valuable comments. Assumption 1 involves Restricted Strong Convexity (RSC) and Restricted Strong Smoothness (RSS), which are properties typically assumed in high-dimensional statistical theory. They imply that the objective function behaves like a strongly convex and smooth function over a sparse domain, even if the function itself is non-convex. Thus, assumption 1 is much weaker than the general strongly convex and smooth condition, which is common and general in the study of $\ell_0$ problems. This problem itself is NP-hard, and despite the flaws in our assumption, we can still gain some insights from it.
>
> In adversarial settings, the objective often involves crafting inputs that cause the network to misclassify. This process can be viewed through the lens of optimizing a loss function that characterizes the discrepancy between the current output and the desired adversarial outcome. The assumption 1 is more relevant to optimization problems, particularly in specific subsets of the domain, and is useful if the standard convexity may not hold. Thus, assumption 1 is relatively plausible in the adversarial attacks.
>
> Q3: Does theorem 2 provide a meaningful performance bound of the algorithm 1 in practice?
>
> A3: Thank you for your insight regarding the tightness of the bound and the role of $\rho$. First, Theorem 2's guarantees are based on two critical assumptions: RSC, RSS, and the boundedness of the function $f(x_0+\delta, y)$. The theorem establishes a geometric convergence rate for Algorithm 1, where $\rho$ is a critical factor. The concern is that $\rho$ might be greater than 1 and hence could cause the bound to diverge as $T \rightarrow \infty$ is valid in a theoretical sense. In practical scenarios, $\rho$ is a factor derived from the assumptions and the structure of the algorithm. It is not an arbitrary variable but is grounded in the algorithm's design and the conditions under which it operates. We acknowledge that the condition of $\rho\leq1$ is critical for the convergence guarantee. Thus, we provide further analysis at Remark 1, where if you set $k=\frac{L^2_{k+k^*}}{\alpha^2_{k+k^*}}k^*+\hat{k}$, then $\rho=1-\frac{\alpha^2}{L^2}\leq 0$. In other words, we can make $\rho \leq 1$ by taking the value of k.
>
> Besides, it is worth emphasizing the error term on the right-hand side of the inequality, which comes from two parts. In the first part, the subproblem of each iteration is NP-hard and solved by a heuristic algorithm, which brings systematic error. The second part is the error caused by zeroth-order gradient estimation. Considering the problem itself is NP-hard, we proved the linear convergence of the algorithm under the condition that the system error is achieved. For the result of convergence, we referred to $\ell_0$ [1] and group $\ell_0$ [2], and we obtained the same result. Although not perfect, it can be accepted in sparse optimization under NP-hard conditions.
>
> [1] de Vazelhes, William, et al. "Zeroth-Order Hard-Thresholding: Gradient Error vs. Expansivity." *Advances in Neural Information Processing Systems* 35 (2022): 22589-22601.
>
> [2] Jain, Prateek, Nikhil Rao, and Inderjit S. Dhillon. "Structured sparse regression via greedy hard thresholding." *Advances in neural information processing systems* 29 (2016).

---

### Official Review · Reviewer_APQW · 2023-10-31

**Soundness:** 3 good
**Presentation:** 4 excellent
**Contribution:** 2 fair
**Rating:** 5
**Confidence:** 4

**Summary:**

The paper considers a novel attack setting, namely, region-based attack that lies in between the pixel-wise and the global attacks and allows additional interpretability of the final perturbation. They evaluate their attack on different datasets such as ImageNet, MNIST and CIFAR and compare to existing methods such as AutoZoom and Square attack etc with respect to numerous performance metrics.

**Strengths:**

-	The paper considers an interesting approach to crafting black-box adversarial examples and compare it to existing paradigms with respect to different aspects.

-	The theoretical background of the method seems to be reasonable

**Weaknesses:**

-	The proposed method is outperformed on ImageNet by existing Square Attack (Table 14 in Appendix D.3). Please include this table in the main part of the paper (at least in some reduced form) or comment on this aspect explicitly. Knowing comparative performance on high-resolution images is important for the readers to get full picture of the method's strengths and weaknesses.

-	Please add median number of queries for your ImageNet experiments in Table 14 as you did for CIFAR10 and MNIST in Tables 3, 4. Just the average number of queries is not sufficient in my opinion.

-	Figure 5 is rather misleading because for the Square Attack in the second row we only see stripe initializaton without any sampled squares. If it is an adversarial example, then it has fooled the image with a single query. It doesn’t provide a good visual impression of what a Square Attack perturbation typically looks like.

**Questions:**

- Could you elaborate on why fixed versions of existing attacks (e. g. Fixed-ZO-NGD) would be valuable baselines? The attacks were not designed that way and introducing this additional constraint seems to be an unclear step to me.
- Why would considering $\ell_{\infty}$ and $\ell_2$ metrics simultaneously e. g. in Table 2 be significant? If we wanted to minimize them simultaneously with the baseline attacks that you consider, we could include it as another term in the loss that they are trying to optimize. Have you considered such modifications to obtain better baselines?

---

> ### Author Response · Authors · 2023-11-21
>
> A1: Thank you for your comments. We have modified the structure of the article according to your suggestion and added the Median item in Tab. 15, and Tab.16 in the appendix.
>
> A2: Thank you for your suggestion. We have amended the figure in Fig. 5 of the main paper.
>
> A3: In order to preserve the inherent characteristics of the baseline algorithm, we have made conservative changes to the existing algorithm (i.e. fixed region), and provided attack results for other areas in the appendix to create as fair a comparison environment as possible. In addition, we added a fixed region experiment. First, we randomly generated 5 squares with the same size on the image and fixed the 5 positions to attack. In order to ensure the consistency of the experiment, we ensured that these 5 groups disturbed 10% of the dimension of the whole image. The result is shown in F.5 Table 22 of the appendix. The visual adversarial sample is shown in Fig. 12 of the appendix.
>
> As far as we know, in recent years, there has been a notable lack of state-of-the-art (SOTA) black-box regional attack algorithms under the same scenario setting, so we changed the SOTA global attack algorithm to a fixed version. The good thing about this is, first of all, fixed versions of existing attacks provide a valuable baseline as they allow for a direct comparison under similar constraints. Secondly, it also effectively measures the attack performance of SOTA global attacks with the perturbation quantity limitation. As can be seen from Tab. 4 in the main part of the paper, both the attack success rate and the query efficiency show a significant decline after the disturbance quantity limitation is added, thus highlighting the challenge of our paper.  Besides, from the results of the five positions provided in the appendix, it can be seen that most of the best performance is concentrated in the center area, but it is still far behind us.
>
> By comparing against these adapted baselines, the unique contributions and advantages of our method become more apparent. This includes demonstrating how automatic region selection can outperform traditional methods that do not focus on region perturbations. While these attacks were not originally designed with fixed constraints, adapting them in this manner helps to create a more comprehensive and robust evaluation framework. This approach is not intended to undermine the original design of these attacks but rather to provide a clearer context for evaluating the specific advancements our method offers.
>
> A4: The goal of our method is to generate sparse and imperceptible perturbations. Therefore, we designed two constraints i.e. $\ell_\infty$ and $\ell^G_0$, where $\ell^G_0$ is the group sparsity and $\ell_\infty$ is the constraint that the control perturbation is not perceptible. The $\ell_2$ norm is a metric to measure the imperceptibility of perturbations outside the constraint.
>
> According to the original characteristics of the baseline, we divided the experiment into three sets(Global attack, Region-wise attack, and Fixed attack).
>
> * In the global attack, all the baseline algorithms have only $\ell_\infty$ constraints, so the sparsity and undetectability of perturbations are measured by $\ell_0$ and $\ell_2$ metrics, respectively.
> * In the region-wise attack, all the baseline algorithms have only $\ell_0$ constraints, so the undetectability of perturbations is measured by $\ell_\infty$ and $\ell_2$ metrics.
> * In the fixed attack, all the baseline algorithms have $\ell_{0+\infty}$  constraint, thus, it just needs to compare the ASR and query efficiency.
>
> A5: We introduce an $\ell_2$ norm-based loss term into the optimization problem. This addition aims to refine the adversarial perturbations by considering the Euclidean distance in the perturbation space. We utilize a standard image dataset CIFAR10 for evaluating the adversarial attacks. We set $\ell_\infty$ constraint as 0.1, and perturb 10\% pixels. We set the parameter of the penalty term, i.e. $c$ is 0.01, 0.1, 0.5, 1, 5, 10. As shown in Tab. 23 of the appendix. As $c$ increases, the $\ell_2$ decreases.

---

> ### Comment · Reviewer_APQW · 2023-11-22
> **Response**
>
> Thank you for your response.
>
> A1. I see that you have included region-wise attacks on ImageNet in the main part. However, I have not noticed any reference to global attack results on ImageNet, namely, Table 15 in the Appendix. Could you at least add a sentence summarizing the results of Table 15 to the main part? I think that it could be relevant for the readers to know that baseline Square Attack requires orders of magnitude fewer queries than your attack in the global setting while having the same ASR.
>
> A3. You write:
>
> "As far as we know, in recent years, there has been a notable lack of state-of-the-art (SOTA) black-box regional attack algorithms under the same scenario setting, so we changed the SOTA global attack algorithm to a fixed version."
>
> Could you provide references that point out that there was a notable lack of such attack algorithms? As far as I know, there was quite a lot of work on, for example, adversarial patch attacks that seem to fall under this category.
>
> A3. I see that, for example, Fixed-Square is significantly weaker than Square (Table 2 vs Table 4). I agree that sometimes constructing an artificial baseline makes sense when there are no suitable existing baselines. But in case of regional attacks Sparse-RS for adversarial patches can serve as such baseline e. g. in Table 4. I would find the comparison to Sparse-RS for patches more suitable than comparing to your own modifications of existing global attacks.
>
> A5. What I meant was not introducing $\ell_2$ constraint into your loss but introducing it into the losses optimized by baseline attacks to which you compare. Since you already compare to your own modifications of existing attacks (e. g. Fixed-* versions), one other way to obtain reasonable baselines would be to, for example, add the $\ell_2$-norm of the perturbation into e. g. Square Attack. The optimization objective can be modified quite easily. That is just the value that we compute after each step and compare to the previous value. I made this suggestion because you compare to existing attacks in terms of both $\ell_\infty$ and $\ell_2$ perturbation norms. Since your method optimizes both norms simultaneously, in my opinion, it would be fair to compare to a baseline that does it as well.

---

> > ### Author Response · Authors · 2023-11-23
> >
> > A1: Thank you for your correction. We have added the discussion on the effect of the global attack algorithm in the corresponding main text. You can also obtain the discussion as follow.
> >
> > In a comparative study with global attack algorithms, Square Attack and Our$_{100\%d}$ both achieve a 100\% success rate in image manipulation, highlighting their effectiveness. However, the Parsimonious algorithm, though effective on the ViT model, is less so on Inceptionv3, suggesting limited model compatibility for ZO-NGD and Parsimonious in constrained query scenarios. Our 30\% results indicate that a 70\% reduction in perturbation leads to a minor decrease in success rate, a reasonable trade-off given stricter constraints require more queries. Our approach and Square Attack show similar success rates and query numbers under identical constraints. In more stringent scenarios, our method effectively balances reduced pixel disturbance with a slight reduction in success rate, maintaining a reasonable query count.
> >
> > A3: Thank you for your valuable comments. We appreciate the opportunity to clarify our position. Regarding the notable lack of zone-based attack algorithms with similar constraints to ours, we offer the following explanation:
> >
> > Our research proposes an automatic region selection attack in a black-box setting, aiming to create disturbances that are both sparse and imperceptible. We achieve sparsity through group constraints and control imperceptibility via $\ell_\infty$ constraints. Within this specific scenario, finding a region attack algorithm that shares these constraints has been challenging.
> >
> > Firstly, traditional adversarial patch attacks are generally noticeable to the human eye. While these attacks are indeed regional, our focus diverges, especially in terms of automatic region selection in black-box environments. Typical patch attacks involve placing conspicuous patches in the image, such as in Sparse-RS[1]. These perturbations are spatially continuous and often do not limit the perturbation magnitude per pixel [4], making the patches easily detectable.
> >
> > Secondly, our motivation is different. Many strategies against patch attacks prioritize the naturalness and portability of patches, incorporating additional features like textures and physical stickers for real-world applicability, as seen in PatchAttack[2], TNT Attack[3], and Adversarial Patch Attack[5]. In contrast, our aim is to explore fragile image regions within constraints to produce sparse yet imperceptible perturbations. This leads us to establish our own criteria for determining the success of adversarial samples.
> >
> > In addition, Location-optimization[6] proposed to jointly optimize the location and content of an adversarial patch, but it belongs to the white-box attack for image classification, which needs to know detailed information of target classifiers.
> >
> > Given these differences in experimental scenarios and the dynamics of patch attacks, our focus was not primarily on adversarial patch attacks. Nevertheless, recoggnizing that adversarial patch attacks fall under the broader category of region-wise attacks, we included Patch-RS in Sparse-RS, a recent advancement in patch attack methodologies, as a reference in our work. We appreciate your insights, which have helped us refine our perspective and approach. Thank you for enriching our discussion.
> >
> > [1] Croce, Francesco, et al. "Sparse-rs: a versatile framework for query-efficient sparse black-box adversarial attacks."  *Proceedings of the AAAI Conference on Artificial Intelligence* . Vol. 36. No. 6. 2022.
> >
> > [2] C. Yang, A. Kortylewski, C. Xie, Y. Cao, and A. Yuille, “Patchattack: A black-box texture-based attack with reinforcement learning,” in ECCV, 2020, pp. 681–698.
> >
> > [3] Doan, Bao Gia, et al. "Tnt attacks! universal naturalistic adversarial patches against deep neural network systems." *IEEE Transactions on Information Forensics and Security* 17 (2022): 3816-3830.
> >
> > [4] Wei, Xingxing, et al. "Simultaneously optimizing perturbations and positions for black-box adversarial patch attacks." *IEEE transactions on pattern analysis and machine intelligence* (2022).
> >
> > [5] Tom B Brown, Dandelion Mané, Aurko Roy, Martín Abadi, and Justin Gilmer. Adversarial patch. arXiv preprint arXiv:1712.09665, 2017.
> >
> > [6] S. Rao, D. Stutz, and B. Schiele, “Adversarial training against location-optimized adversarial patches,” in ECCV, 2020, pp. 429–448.

---

> > > ### Author Response · Authors · 2023-11-23
> > >
> > > A5: Thank you for your suggestion, we added a $\ell_2$ loss term in the global attack algorithms. The result can be obtained in the following table.
> > >
> > > We randomly select 50 samples on ImageNet for untarget attack. We do the attack on two models Inceptionv3 and ViT-B/16, where the penalty coefficient $\eta$ of $\ell_2$ loss is 0.1, 1,10, respectively. We can see that as $\eta$ increases, the $\ell_2$ distance decreases, the success rate also loses, and the number of queries increases. This result is intuitive. It is worth mentioning that after we add the loss term to Square Attack, none of these samples can be successfully attacked, from which we can see that Square Attack cannot deal with such a problem setting. ZO-NGD and Parsimonious attacks can obtain the corresponding results. From this result, we can see that a larger $\eta$ is needed to achieve the effect of the constraint through the $\ell_2$ loss term. The parameter $\eta$ needs to be more carefully tuned.

---

> ### Author Response · Authors · 2023-11-23
>
> | Algorithm                 | Model        | $\eta$ |              ASR |      Avg. queries |   Med. Queries | $\ell_0$ |      $\ell_2$ |
> | :------------------------ | :----------- | -------: | ---------------: | ----------------: | -------------: | ---------: | --------------: |
> | Square                    | ViT-B/16     |      0.1 |                0 |             20000 |          20000 |     147391 |            23.6 |
> |                           |              |        1 |                0 |             20000 |          20000 |     147437 |           18.85 |
> |                           |              |       10 |                0 |             20000 |          20000 |     147285 |  **17.9** |
> |                           | Inception-v3 |      0.1 |                0 |             20000 |          20000 |     147214 |           19.86 |
> |                           |              |        1 |                0 |             20000 |          20000 |     147447 |            19.1 |
> |                           |              |       10 |                0 |             20000 |          20000 |     147238 | **17.11** |
> | ZO_NGD                   | ViT-B/16     |      0.1 | **0.9426** | **2750.36** |  **400** |     133254 |           29.08 |
> |                           |              |        1 |           0.9159 |           2938.85 |  **400** |     142833 |           29.01 |
> |                           |              |       10 |           0.8079 |           6104.02 |            800 |     145268 |  **26.3** |
> |                           | Inception-v3 |      0.1 | **0.6604** | **8118.74** | **1400** |     253082 |            29.2 |
> |                           |              |        1 |           0.6469 |           8956.36 |           5200 |     243126 |           28.67 |
> |                           |              |       10 |           0.6347 |           9036.54 |           6400 |     245871 | **28.41** |
> | Parsimonious              | ViT-B/16     |      0.1 | **0.9231** | **2054.69** |  **440** |     147251 |           37.33 |
> |                           |              |        1 |           0.8974 |           2708.05 |            540 |     147187 |           37.27 |
> |                           |              |       10 |           0.7949 |           4908.94 |            695 |     147225 | **37.21** |
> |                           | Inception-v3 |      0.1 | **0.9697** | **1260.27** |  **446** |     261955 |           49.67 |
> |                           |              |        1 |           0.9197 |           1437.06 |            555 |     264928 |           49.61 |
> |                           |              |       10 |           0.8697 |           1873.39 |            507 |     261936 |  **49.5** |
> | Our$_{Non-overlapping}$ | ViT-B/16     |      0.1 | **0.9568** |            984.16 |            222 |     142440 |           17.59 |
> |                           |              |        1 |           0.9468 |  **959.27** |            216 |     144000 |           17.64 |
> |                           |              |       10 |           0.9309 |           1048.83 |  **192** |     142893 | **17.63** |
> |                           | Inception-v3 |      0.1 |   **0.94** |  **632.83** |            234 |     264098 |           43.87 |
> |                           |              |        1 |           0.9334 |            895.47 |  **218** |     262990 |           43.56 |
> |                           |              |       10 |           0.9307 |            907.57 |            219 |     263672 | **42.05** |
> | Our$_{Overlapping}$     | ViT-B/16     |      0.1 | **0.9454** |  **689.04** |  **131** |     132572 |           20.16 |
> |                           |              |        1 |           0.9353 |            763.03 |            132 |     142322 |           20.14 |
> |                           |              |       10 |           0.9254 |           1162.89 |            156 |     132356 | **19.98** |
> |                           | Inception-v3 |      0.1 |            0.927 |  **727.63** |   **97** |     253276 |           44.84 |
> |                           |              |        1 | **0.9286** |            818.64 |            204 |     253073 |           44.67 |
> |                           |              |       10 |           0.9163 |           1164.36 |            136 |     244644 | **43.08** |

---

> > ### Comment · Reviewer_APQW · 2023-12-04
> > **Response**
> >
> > I once again thank the authors for very detailed replies. I am increasing my overall score

---

### Official Review · Reviewer_jyb5 · 2023-11-04

**Soundness:** 3 good
**Presentation:** 3 good
**Contribution:** 2 fair
**Rating:** 5
**Confidence:** 4

**Summary:**

This paper provides a region-wise black-box attack method. It automatically identifies relevant regions based on a dependable standard, rather than relying on fixed regions or heuristics. It treats the problem as an optimization problem with technical constraints, namely lG0 and l∞. lG0 represents structured sparsity defined within a specific collection of groups G, enabling the automatic detection of regions requiring perturbation. The optimization is solved using a natural evolution strategies algorithm. It also discusses the group overlapping separately. When G doesn’t overlap, the author uses a closed-form solution for the first-order Taylor. When G overlaps, it adopts an approximate solution by using a greedy selection on G. The convergence of the algorithm is also discussed. In the experiments, the authors compare its performance with global region and pixel-wise attack models. Demonstrating good performances compared with baseline methods.

**Strengths:**

+ The authors provided a convergency analysis of the attack methods, which would be useful when a guarantee of the model robustness is needed.
+ Experiments have been done on large-scale image datasets, and comparison has been done with recent black-box attack methods.

**Weaknesses:**

- Some descriptions are confusing. For example, at the beginning of sec. 2.2,  '... we propose a new objective function for adversarial training ...' while it should be 'adversarial attack'? Also, there are many notations that appear without definition. Like what is I_G in Theorem 1? Besides, the deduction in sec. 3.1 seems to be unnecessary, and the gradient estimation proposed in equ. (4) does not seem to differ from the standard gradient estimation approach.
- For computational cost and convergence, it only says high, medium and low. Are there any quantitative results to demonstrate it?
- It is not clear how the algorithm performs region selection. In algorithm 1, how the delta^0 is initialised? What is the initial perturbation group set G?
- Using the estimated gradient to perform black-box adversarial attacks is not new. Please refer to the SPSA attack [1].
- In the experiment section. I am not sure if the comparison is fair, as different black-box attacks select different regions. Besides, the result shows that the proposed methods may actually require more queries as the median is significantly higher than other models. Also, the authors used a simple CNN for the CIFAR10 dataset. It would be more convincing to evaluate pre-trained models from PyTorch model zoo or other resources.
- As the authors conducted a convergence analysis of the proposed attack. I am wondering if this can be further developed towards an adversarial verification method. Also, the robustness verification of adversarial patches has been done in [2]. I am also interested in the performance of the proposed attack on such a certified defence.

[1] Uesato, Jonathan, et al. "Adversarial risk and the dangers of evaluating against weak attacks." ICML, 2018.

[2] Salman, Hadi, et al. "Certified patch robustness via smoothed vision transformers." CVPR, 2022.

**Questions:**

Pls see the Section Weaknesses.

---

> ### Author Response · Authors · 2023-11-21
>
> Q1: Can the authors clarify the inconsistencies in the manuscript, particularly the mislabeling in Section 2.2, undefined notations such as 'I_G' in Theorem 1, and the need for the deduction and gradient estimation method in Section 3.1 and Equation (4)?
>
> A1: We appreciate the opportunity to clarify the inconsistencies you have pointed out in our manuscript.  Firstly, we have fixed mislabeling in Section 2.2 and undefined notations in Theorem 1. Additionally, we have shortened the content of this section 3.1. and the gradient estimation method in Equation (4).  Significantly, we integrate this estimation technique with the Natural Evolutionary Strategies (NES). This integration is key to efficiently navigating the complex search space inherent in our proposed method.
>
> As for the gradient estimation in Equation (4), while it shares similarities with standard approaches, it incorporates unique adaptations for our specific problem setting.  These include modifications to account for structured sparsity and $\ell_\infty$ constraints in the black-box adversarial context.
>
> Q2: For computational cost and convergence, it only says high, medium, and low. Are there any quantitative results to demonstrate it?
>
> A2: Thank you for your feedback regarding the presentation of computational cost and convergence in our paper.
>
> In Table 1 in the main part of the paper, we opted for qualitative descriptors (high, medium, low) to provide an initial, high-level comparative overview of our approach against others. This was to give the reader a quick reference to the differences between different attack patterns. We acknowledge that this presentation is flawed, and we have revised the table in the main text so that readers can better understand the different attack patterns. The modified version is shown as below.
>
> |    Attack Type    |          Description          | Visibility |              Objective              |
> | :---------------: | :----------------------------: | :--------: | :---------------------------------: |
> |      Global      | Alters entire image uniformly. |   Highly   |   Degrade overall image quality.   |
> |     Regional     |    Targets specific areas.    | Moderately |  Conceal or alter specific parts.  |
> | Sparse Pixel-wise | Alters a few scattered pixels. |   Least   | Sparse and conspicuous disturbance. |
>
> For cost analysis, we provide quantified results for reference in Appendix B.3.
>
> Table 6: The complexity of optimizing the attack with the current region.
>
> |      Lines      | Time Complexity  |
> | :-------------: | ---------------- |
> |       3-6       | $O(nd)$        |
> |        7        | $O(n)$         |
> |       8-9       | $O(d)$         |
> |       12       | Region Selection |
> |       16       | $O(d)$         |
> |       17       | $O(1)$         |
> | **Total** | $O(Tnd)$       |
>
> Table 7: The complexity of the region selection process.
>
> | Lines           | Time Complexity |
> | --------------- | --------------- |
> | 3               | $O(d)$        |
> | 4               | $O(d)$        |
> | 5               | $O(d)$        |
> | 6               | $O(d)$        |
> | 7               | $O(d)$        |
> | **Total** | $O(kd)$       |
>
> Q3: It is not clear how the algorithm performs region selection. In algorithm 1, how the delta^0 is initialized? What is the initial perturbation group set G?
>
> A3: Thank you for your valuable feedback on the clarity of our algorithm's region selection process and the initialization details in Algorithm 1.
>
> Before the attack starts, we group the dimensions of the images according to preset criteria, and the mask generated by the grouping is independent of the image, so this saves a lot of overhead. Our grouping method is to divide groups by a fixed sliding window. It can generate different perturbations by adjusting the window size and step size. In section 3.3, we explain the process of the algorithm in detail, where the 9 line calculates the **DIS** of all groups according to the mask $I_G$. Then, k groups with the smallest **DIS** are selected by algorithm 2. The selection process is iterative and adaptive, meaning it can change in subsequent iterations based on the feedback received from the model's responses to previous perturbations.
>
> In the version provided in the paper, we set the initial value of $\delta$ to be a uniformly distributed random disturbance. The initial perturbation group set G is empty. There are many hot-start initializations, such as calculating DIS after random noise and then selecting k groups heuristically, or like Square attack initializations, initializing on a disturbed boundary and then selecting k groups based on **DIS**. All hot boot methods are available in the code.

---

> ### Author Response · Authors · 2023-11-22
>
> Q4: Using the estimated gradient to perform black-box adversarial attacks is not new. Please refer to the SPSA attack [1].
>
> A4: Thank you for pointing out the relevance of the SPSA attack in the context of our work. Our work contributes to the field by extending the concept of gradient estimation to more specific and challenging scenarios of adversarial attacks. So black box gradient attack method is not our main work and innovation point. Our method specifically addresses the challenge of region-wise adversarial attacks, focusing on improving efficiency and imperceptibility in this narrower domain. We propose specific algorithmic modifications and enhancements that are tailored to the unique requirements of region-wise attacks. This includes adaptations to handle structured sparsity constraints and the $\ell_\infty$ norm in a black-box setting. Our approach integrates gradient estimation with Natural Evolutionary Strategies (NES), creating a novel synergy that enhances the efficacy of our attack method in terms of query efficiency and perturbation control.
>
> Q5: Is the experimental comparison fair considering the different regions selected by various black-box attacks?
>
> A5: As far as we know, in recent years, there has been a notable lack of state-of-the-art (SOTA) black-box regional attack algorithms under the same scenario setting, so we changed the SOTA global attack algorithm to the fixed version.  The good thing about this is, first of all, fixed versions of existing attacks provide a valuable baseline as they allow for a direct comparison under similar constraints.  Secondly, it also effectively measures the attack performance of SOTA global attacks with the perturbation quantity limitation. In the appendix of the paper, we also added the attack performance of the global attack algorithm on other fixed regions. In addition, we added a new set of experiments. First, we randomly generated 5 squares with the same size on the image and fixed the 5 positions to attack. In order to ensure the consistency of the experiment, we ensured that these 5 groups disturbed 10% of the dimension of the whole image. The result is shown in F.5 Table 22 of the appendix. The visual adversarial sample is shown in Fig. 12 of the appendix.
>
> It is worth noting that the novelty of our method is precisely the automatic region selection, which is the key difference from other methods. The experimental setup was designed to demonstrate this novelty and its impact on the overall performance of the attack. In the face of heuristic Patch-RS attacks and global attacks with different characteristics, we need to pay attention to the constraints and background of the problem. As for how to select the region, we believe that the inherent characteristics of other algorithms should be retained. For the global algorithm, we made conservative modifications and provided attack results of multiple different regions, as well as adding experiments of multiple sub-regions to evaluate all algorithms comprehensively.
>
> Q6: Also, the authors used a simple CNN for the CIFAR10 dataset. It would be more convincing to evaluate pre-trained models from PyTorch model zoo or other resources.
>
> A6: We evaluated the CIFAR10 dataset on two models, resnet18 and mobilenet_v2. The results of the evaluation are shown in the Tab. 21 of the appendix. In this evaluation, we also adopted variance reduction technology, and its query efficiency and attack success rate have been effectively improved.
>
> In Resnet18, untargeted attacks had higher success rates in non-overlapping conditions, while targeted attacks were more successful in overlapping conditions. For Mobilenet-v2, untargeted attacks showed slightly better performance in non-overlapping conditions, but targeted attacks were more effective in non-overlapping scenarios. Overall, Mobilenet-v2 required more queries than Resnet18, indicating lower query efficiency. The $\ell_2$  norm values remained consistently around 1.7 for both models, suggesting similar levels of perturbation. This suggests that variance reduction technology improved attack success rates and had a notable impact on query efficiency.

---

> ### Author Response · Authors · 2023-11-22
>
> Q7:  Can the convergence analysis lead to an adversarial verification method, and how effective is the proposed attack against certified defenses like those in Salman et al.'s (2022) work on smoothed vision transformers?
>
> Thank you for your comments. We trained a robust model on CIFAR10 datasets, the Vision Transformer model. We conducted an experiment with Patch-RS attack on the CIFAR10 dataset for untargeted attacks because we are all region-wise attacks. We randomly selected 200 samples and set the query budget to 10,000. In order to maintain the consistency of the experiment, we followed the Patch-RS algorithm setting. We set the $\ell_\infty$ constraint to 1, and perturb a maximum of 500 dimensions. Confronting such a robust model for the patch, the performance between ours and Patch-RS is shown in the following table. As can be seen from the following table, our algorithm is more robust than the heuristic Patch-RS algorithm.
>
> | Algorithm | ASR    | Avg. queries | Med. queries |
> | --------- | ------ | ------------ | ------------ |
> | Patch-RS  | 79.57% | 2284.70      | 110          |
> | OURS      | 81.72% | 2032.82      | 4            |

---

### Official Review · Reviewer_PV33 · 2023-11-09

**Soundness:** 3 good
**Presentation:** 4 excellent
**Contribution:** 3 good
**Rating:** 6
**Confidence:** 5

**Summary:**

This papers focus on black-box adversarial attacks where the objective is to construct strong adversarial perturbations with only query-access to the black box deep neural network. The novelty of the proposed work lies in the automatic region selection approach based on natural evolution strategy, which can even be derived as a closed-form solution for non-overlapping patches. It further demonstrates the success of the proposed attack on mnist, cifar 10, and imagenet dataset.

**Strengths:**

This paper is very well written and the evaluation pipeline is rigorous. Authors have evaluated the attack's strength both analytically and empirically across three datasets and multiple different ablations.

**Weaknesses:**

Query efficiency: I couldn’t find the comparison on how efficient is the current attack w.r.t the previous attacks. When permitted a high number of queries, the strength of most black-box attacks would increase, thus making it an unfair comparison in table 2.

Second, it is critical to provide the number of queries vs attack strength to identify the pareto optimal curve of the current attack (currently the number of queries are set to fixed 10k, 40k - not sure why?). Similarly it is necessary to compare queries vs ASR plot with other attacks.

Are there diminishing returns in attack success with higher resolution? While the proposed attack appears to be stronger and less perceptible than baselines of small resolution datasets (cifar10, mnist), the trend doesn’t fully hold on ImageNet dataset (table 14 in appendix). Square attack [1] achieves equally high success rate and lower average queries.

1. Andriushchenko, Maksym, Francesco Croce, Nicolas Flammarion, and Matthias Hein. "Square attack: a query-efficient black-box adversarial attack via random search." In European conference on computer vision, pp. 484-501. Cham: Springer International Publishing, 2020.

**Questions:**

Can authors clarify how the attack complexity behaves with experimental setup, i.e., input resolutions, size of neural networks, number of classes, etc?

Can authors provide additional intuition of why the proposed approach has higher ASR than similar black-box attacks, e.g., square attacks? Is it because of attack strength or subtle design choices, such as patch based perturbations.

---

> ### Author Response · Authors · 2023-11-21
>
> Q1: How does the efficiency of the current attack compare to previous ones, considering the high number of queries allowed, which may lead to an unfair comparison in Table 2?
>
> A1: Thanks for your valuable comments. Our experiments were designed with a focus on query efficiency, recognizing the real-world cost and resource implications of querying models. This strategic approach ensures that our method's performance is rigorously assessed within realistic operational constraints. As you said, augmenting query budgets does indeed amplify the attack intensity. The comparison in Table 2 was designed to demonstrate the efficiency of our method under a specific set of conditions. We compared the performance of all algorithms with the same limited query budget. As far as we know in the baseline, the setting of this query budget is not very strict, and can effectively compare the differences between different algorithms.
>
> Our study presents a notable comparison between our method and global attack algorithms, demonstrating comparable Attack Success Rate (ASR) and an average of queries, yet with enhanced performance under the dual constraints of limited query budgets and $\ell_0$ constraint. To preserve the inherent characteristics of each baseline, we have conservatively modified SOTA global attack algorithms (fixed attack). These findings are evident in the fixed version experiments, which underscore the efficacy of our methodology. The result can be obtained in Tab.4, Tabl.5, and so on. In order to conduct a more comprehensive comparison with global attack algorithms, we conducted an experiment that fixed multiple regions in each sample to attack. Detailed results can be obtained in the F.5 of the appendix. Fixing multiple areas to attack is more effective than attacking just one area, which also explains the need to explore vulnerable areas of the image. Compared with the heuristic exploration method, we expect to automatically select the vulnerable regions of the image from the perspective of optimization.
>
> Q2: Why is the number of queries fixed at 10k and 40k, and how does this affect identifying the Pareto optimal curve and comparing the queries vs. ASR plot with other attacks?
>
> A2: As mentioned in our previous answer, our algorithm simulates the performance of all attack algorithms under limited queries in reality. Therefore, we set query budgets for different models and data sets according to the consideration of image dimension, model complexity, and other aspects. According to our survey results, this query budget setting is universal. And it can effectively compare the attack strength of different algorithms.
>
> Setting different query budgets can comprehensively measure the performance of attack algorithms under different query budgets. We consider adding the attack intensity of different query budgets to our experimental scenario. Thus, we've added a set of experiments to Appendix F.3. We set different query budget that is 5,000, 10,000, 15,000, 20,000, and 25,000 instead of a fixed budget. Notwithstanding the inherent disparities in our respective problems, we divided three sets of experiments for different problems (Global attack, Region-wise attack, Fixed attack). First, we compared with global attack methods, i.e. Square attack, Parsimonious attack, and ZO-NGD attack, with $\ell_\infty$ constraints in Fig.11(a,b). Thus, we set the same $\ell_\infty$ constraint for all algorithms is 0.05. Second, we compared the region-wise attack, i.e. Patch-RS, with a patch in Fig.11(c,d). We set the patch size of the patch to 80 in Patch-RS. The $\ell_0$ constraint is 19200(80 $\times$ 80 $\times$ 3). Finally, we compared with the fixed-version of global attack algorithms in Fig. 11(e,f). For all algorithms, we set the same double constraint, $\ell_\infty$ is 0.1, $\ell_0$ is 26820 for the ImageNet dataset, Inceptionv3 model, and $\ell_\infty$ is 0.1, $\ell_0$ is 15052 for ImageNet dataset, vision transformer model. That is, for double constraint, we perturb only 10\% of the pixels.
>
> From the Fig. 11(a,b), all algorithms have $\ell_\infty$ constraint 0.05. We can see that the success rate of the attack will decrease significantly if the query is less than 15,000 for high-resolution image attack tasks. When the query budget reaches 20,000, the success rate becomes stable. From the Fig. 11(c,d), all algorithms perturb up to 19200 of the pixels. Patch-RS is a heuristic attack method, it can be seen from the Fig. 11(d) ViT-B model that Patch-RS performs very poorly under a strict query budget. From Fig 11(e,f), global attack algorithms with fixed regions perform poorly under different budgets. Due to time constraints and according to the attack performance of different regions in Tab. 18 and Tab. 19 in the appendix, it can be concluded that most algorithms have the best attack effect at the center of the image, so we provide the results of the fixed central region in Fig. 11(e, f).

---

> ### Author Response · Authors · 2023-11-21
>
> Q3: Is there a trend of diminishing returns in attack success at higher resolutions, as indicated by the weaker performance on ImageNet compared to CIFAR10 and MNIST, and how does this compare to the Square Attack's high success rate with fewer queries?
>
> A3: High-resolution images often contain more complex and varied features, which might affect the efficiency and success rate of adversarial attacks compared to smaller-resolution datasets. As shown in Tab. 15 in the appendix, the attack performance trend observed on CIFAR10 and MNIST does not fully extend to the ImageNet dataset. In the latest results provided (Tab. 5 in the main part of the paper), we adopt the variance reduction method to estimate the gradient in the black box more effectively.
>
> The problem addressed within the realms of the Square attack substantially diverges from the context elucidated within our paper. In our paper, our deliberations pivot around the conceptualization of regionally sparse attacks, a paradigm wherein we endow the capability to autonomously designate target regions for assault while concurrently regulating the number of regions to be subjected to such perturbation.  In stark contrast, the Square attack methodology is fundamentally engrossed in the task of imbuing perturbations across the entire expanse of an image, bereft of the nuanced capacity to pinpoint specific regions meriting subjection to attack.
> From Tab. 15 in the appendix, if we perturb all the pixels like the Square attack, we can achieve similar success rates and queries to the Square attack. However,  from Tab.17, Tab. 18, and Tab. 19 in the appendix, we can see that Square attack doesn't do well with double constraint scenarios.
>
> Q4: Can authors clarify how the attack complexity behaves with the experimental setup, i.e., input resolutions, size of neural networks, number of classes, etc?
>
> A4: Our experimental scenario is set up as an unknown black box attack, and we get no information about the model other than output. This is a big challenge for gradient-based attack algorithms, and we want to generate sparse and imperceptible perturbations, and the strict requirements make the number of queries significantly higher.
>
> We tested our method on three datasets: MNIST, CIFAR10, and ImageNet, each varying in image resolution and complexity. Complex models are also harder to attack than simple CNN networks on CIFAR10 and MNIST datasets. Our evaluation models include complex models(Inception-v3, ViT-B) on high-resolution images. The dimensions of Inception-v3 model images are 268203($299\times299\times3$) and the dimensions of ViT-B model images are 150528($224\times224\times3$). Higher dimensions of the image will cause Higher queries and a lower attack success rate. In addition, there are 1000 classes in the ImageNet dataset, which makes attacks more difficult and needs more queries.
>
> In addition, we provide a simple experimental result to help illustrate the effect of images on the attack algorithm. We set the max query budget is 5,000, and randomly select 300 samples from CIFAR10 and MNIST, randomly select 50 samples from ImageNet. The following table shows the attacks of non-target attacks and non-overlapping groups.
>
> | Dataset  | Model       | Resolution              | ASR    | Avg. queries | Med. queries |
> | -------- | ----------- | ----------------------- | ------ | ------------ | ------------ |
> | MNIST    | CNN         | $28\times28\times1$   | 100%   | 162.75       | 94           |
> | CIFAR10  | CNN         | $32\times32\times3$   | 98.15% | 364.39       | 4            |
> | ImageNet | Inceptionv3 | $299\times299\times3$ | 92.86% | 923.96       | 1            |
> | ImageNet | ViT-B/16    | $224\times224\times3$ | 89.57% | 1258.07      | 115          |
>
> In the follow-up work, we adopted the variance reduction technology to effectively improve the query efficiency. Although our work still has some shortcomings, it also provides the direction for an automatic region selection attack algorithm.

---

> ### Author Response · Authors · 2023-11-22
>
> Q5: Can authors provide additional intuition of why the proposed approach has higher ASR than similar black-box attacks, e.g., square attacks? Is it because of attack strength or subtle design choices, such as patch-based perturbations?
>
> A5: First, our approach utilizes automatic region selection, enabled by the $\ell_0^G$ constraint, which represents structured sparsity defined on a collection of groups G. This automatic detection of the regions that need to be perturbed is a key differentiator from heuristic-based approaches like square attacks. It allows for more targeted and effective perturbations, increasing the ASR.
> Second, we reformulated the function $f$ as an optimization problem with $\ell_0^G$ and $\ell_\infty$ constraint. This formulation, combined with the use of natural evolution strategies and search gradients, provides a robust framework for generating perturbations that are more aligned with the target, contributing to a higher ASR.
> Finally, for non-overlapping groups G, our method provided a closed-form solution to the first-order Taylor approximation of the objective function. In cases where G is overlapping, an approximate solution is derived. Both approaches ensure effective perturbation generation across various scenarios, enhancing the overall ASR.

---

### Author Response · Authors · 2023-11-21

Thanks to all reviewers for your valuable comments and attention to this paper, we have corrected the paper and highlighted it in the PDF file. All added experimental results are concentrated in Appendix F.3. Feel free to ask questions if we need further explanation. Thanks again.

---

### Meta-Review · Area_Chair_dPfJ · 2023-12-09

**Metareview:**

The author propose a black-box adversarial attack where the threat model is to have an $\ell_\infty$-bound together with a bound on the number of perturbed regions. The authors argue that this threat model is more reasonable as a standard $\ell_\infty$-attack or a patch-based attack. The authors design an algorithm, which is to some extent based on gradient estimation, and prove convergence under quite strong conditions.

Strengths:
- theoretical convergence guarantee
- in some cases improvements compared to previous work

Weakness:
- the motivation of the threat model remains unclear. If the attack should be imperceptible then why not just $\ell_\infty$? or if it should be additionaly sparse plus a $\ell_0$-constraint?

- there is no discussion about the assumptions of the convergence results and there is also no experimental evidence of a linear convergence rate. Thus it remains unclear for which kind of classifiers the theorem would hold as one reviewer points out.

- the experimental results are mixed (Square Attack outperforms the proposed method on ImageNet, Table 15) and the evaluation is only done for non-robust models (except one model in the rebuttal) which is insufficient.

In total the reviewers lean towards rejection. I think that the paper needs better motivation why the considered threat model is interesting and then a better discussion of the theoretical results as well as more detailed experimental comparison in particular also for robust models. The reviewers gave a lot of valuable hints how the evaluation could be improved.

Minor comment:
I found the paper not to be very clear, e.g. in Section 3.2 it is said that the Function should be smooth - two lines below one finds out that they mean that the gradient of the function has to be Lipschitz.

**Justification For Why Not Higher Score:**

see above

**Justification For Why Not Lower Score:**

N/A

---

### Decision · Program_Chairs · 2024-01-16

Reject